**Perspective**

# Improving statistical reporting in psychology
Anna-Lena Schubert [1] ✉, Meike Steinhilber [1], Heemin Kang [2] & Daniel S. Quintana[2,3]

Transparent and comprehensive statistical reporting is critical for ensuring the credibility, reproducibility, and interpretability of psychological research. This paper offers a structured set of guidelines for reporting statistical analyses in quantitative psychology, emphasizing clarity at both the planning and results stages. Drawing on established recommendations and emerging best practices, we outline key decisions related to hypothesis formulation, sample size justification, preregistration, outlier and missing data handling, statistical model specification, and the interpretation of inferential outcomes. We address considerations across frequentist and Bayesian frameworks and fixed as well as sequential research designs, including guidance on effect size reporting, equivalence testing, and the appropriate treatment of null results. To facilitate implementation of these recommendations, we provide the Transparent Statistical Reporting in Psychology (TSRP) Checklist that researchers can use to systematically evaluate and improve their statistical reporting practices (https://osf.io/t2zpq/). In addition, we provide a curated list of freely available tools, packages, and functions that researchers can use to implement transparent reporting practices in their own analyses to bridge the gap between theory and practice. To illustrate the practical application of these principles, we provide a side-by-side comparison of insufficient versus best-practice reporting using a hypothetical cognitive psychology study. By adopting transparent reporting standards, researchers can improve the robustness of individual studies and facilitate cumulative scientific progress through more reliable meta-analyses and research syntheses.

Scientific papers are one-way conversations. Given that seeking additional details or clarifications from authors can be challenging and unreliable, transparent and comprehensive statistical reporting is essential for advancing psychological science[1,2]. It allows researchers to demonstrate the rigor and validity of their findings, fostering credibility and facilitating cumulative knowledge building. For authors, clear statistical reporting enhances the clarity of their work, ensuring that readers can fully understand the study's methodology, results, and conclusions. For readers, well-documented statistical details enable critical evaluation of a study's robustness, relevance, and replicability. Moreover, transparent reporting supports meta-analyses and research syntheses, promoting reproducibility and reducing ambiguities.

This paper provides practical guidelines for transparent statistical reporting in empirical psychological research using quantitative methods. Building on the recommendations of the *American Psychological Association* (*APA*)[3] and transparent reporting practices[4,5], we focus on key aspects of quantitative statistical reporting across psychology's diverse subfields. Specifically, we highlight two critical stages where transparency is most impactful: (1) the planning stage (before data collection or analysis) and (2) the data analysis stage. For each stage, we offer guidance to help researchers make informed decisions, identify useful resources, and effectively report key elements. Additionally, we address considerations for different statistical paradigms, including frequentist statistics and Bayesian statistics (see Box 1) and fixed-sample vs. sequential designs (see Box 2).

We cover these statistical approaches because they are widely used in psychology and offer distinct advantages. Frequentist statistics remains the dominant framework, offering concepts such as statistical power and long-term error control. It is typically applied in fixed-sample designs and relies on *p*-values, which are often misinterpreted[6]. In contrast, Bayesian methods offer a probabilistic framework for updating beliefs with data, making them especially useful when prior information is available. In our discussion of Bayesian reporting practices, we focus on hypothesis testing using Bayes Factors (BFs) but also cover approaches focused on posterior estimation.

[1]Department of Psychology, Johannes Gutenberg University Mainz, Mainz, Germany. [2]Department of Psychology, University of Oslo, Oslo, Norway. [3]NevSom, Department of Rare Disorders, Oslo University Hospital, Oslo, Norway. ✉e-mail: anna-lena.schubert@uni-mainz.de

## Box 1 | Different statistical paradigms

**Frequentist statistics:** Frequentist statistics encompasses two distinct statistical theories that have historically been intermingled and often confused, despite their fundamental and incompatible differences. Detailed discussions of this distinction and its implications can be found in both theoretical analyses and pedagogical overviews[221–226]. The term null hypothesis significance testing (NHST) refers to a hybrid of these two frameworks and is therefore deliberately avoided in this paper.

*Significance testing*, as developed by Fisher, follows the principle of inductive inference, where conclusions are drawn from the outcome of a particular experiment[227]. In this approach, a single null hypothesis is specified (e.g., that there are no group differences, or that the group difference is not larger than a specific value), and the goal is to assess how unusual the data are if that null hypothesis is true. Knowledge is gained by rejecting the null hypothesis (the hypothesis is "nullified") based on the magnitude of the discrepancy between the observed results and what the null predicts. The $p$-value is the central statistic in this framework; it quantifies how likely the observed data (or even more extreme outcomes) are, assuming the null hypothesis is true. Researchers may choose different significance thresholds for the $p$-value depending on the context of their study. A small $p$-value suggests that the observed data are unlikely under the null hypothesis, leading to the inference that either a rare event has occurred or that the null hypothesis does not adequately explain the data.

*The theory of statistical decision making*, developed by Neyman and Pearson, follows the principle of inductive behavior[34,228]. In this framework, two exhaustive and mutually exclusive hypotheses are constructed — the null and the alternative hypothesis. The goal is not to draw inferences from a single dataset, but to develop a decision rule that controls the frequency of errors across repeated experiments in the long run. This approach formalizes two types of errors: Type I errors (false positives) and Type II errors (false negatives), and introduces the concept of statistical power — the probability of correctly rejecting the null hypothesis when the alternative is true. Crucially, its purpose is to guide behavior under uncertainty, avoiding the interpretation of single study outcomes as evidence. Decisions are made by setting a fixed criterion (e.g., $\alpha = 0.05$),

collecting a sample based on an a priori power analysis (e.g., $1 - \beta = 0.95$), and determining whether the test statistic falls within the predefined rejection region (e.g., $p \leq 0.05$). The $p$-value can be used as a test statistic in this context, but it is not considered a measure of evidence. If the test statistic falls within the critical region, the null hypothesis is rejected, which implies a decision to act as though the alternative hypothesis is true. However, since it is still unknown whether this conclusion is correct, the decision reflects a rule for behavior under uncertainty, not a confirmation of the truth of the alternative. In this theory, the same experiment is conceptually repeated an infinite number of times, and only under this repetition do long-run error rates and confidence intervals attain their intended interpretations. For example, a 95% confidence interval means that, across many such repeated experiments, 95% of the intervals would contain the true parameter value.

**Bayesian statistics:** Bayesian statistics integrates prior knowledge or beliefs with the observed data through Bayes' theorem to generate posterior distributions that reflect updated beliefs about parameters or hypotheses[47]. Bayesian methods in psychological research can, broadly speaking, serve two complementary purposes. One approach is Bayesian hypothesis testing, most commonly operationalized via Bayes factors, which quantify the relative evidence for $H_1$ versus $H_0$ given the data and the specified priors. For example, a Bayes factor ($BF_{10}$) of 5 indicates that the observed data are five times more likely under $H_1$ than under $H_0$. This approach is often used for model comparison and allows researchers to formally evaluate evidence in favor of the null hypothesis relative to an alternative hypothesis[229].

Another common Bayesian approach is estimation and posterior inference, where the focus lies on characterizing effect sizes and quantifying uncertainty rather than making a dichotomous decision about $H_0$ vs. $H_1$. In this framework, posterior distributions, credible intervals, and posterior probabilities provide a nuanced understanding of the data and the likely range of underlying effects[98]. Bayesian methods thus offer a flexible inferential framework that encompasses both hypothesis testing and parameter estimation, providing richer information than binary significance testing alone.

Beyond the choice of inferential framework, researchers must also consider how data collection is planned. This is where sequential designs come into play. Rather than constituting a separate inferential paradigm, a sequential design is a study-planning strategy that can be implemented within either frequentist or Bayesian frameworks. It enhances efficiency by allowing researchers to stop data collection once pre-specified decision criteria are met. Ultimately, both the inferential framework (frequentist vs. Bayesian) and the data collection strategy (fixed-sample vs. sequential) should be chosen based on the study's goals, the availability of prior knowledge, and practical resource constraints. While the former determines how evidence is quantified and interpreted, the latter dictates when data collection begins and ends.

Our overarching goal is to enhance the credibility and interpretability of research findings. Improved statistical reporting clarifies the theoretical, practical, and societal implications of results by helping ensure accurate effect size estimation. Furthermore, these practices facilitate large-scale meta-analyses and systematic reviews by enabling machine-readable reporting. To achieve this, we present a comprehensive guide to statistical reporting in psychology. We begin with key considerations at the planning stage, including preregistration and sample size determination. Next, we outline best practices for reporting statistical models, effect sizes, and inferential results. Finally, we discuss common errors, strategies for interpreting null results, and approaches for ensuring reproducibility. While this guide does not cover machine learning or qualitative analysis reporting, we

provide additional resources in Box 3 for researchers interested in these methods.

In contrast to many previous guidelines that remain largely conceptual, our goal is also practical. Researchers frequently encounter uncertainty about how to implement best-practice recommendations using the software and tools available to them. For example, they may understand that confidence intervals (CIs) should be reported for ANOVAs but remain unsure how to compute them or which R packages offer this functionality. Throughout the paper we provide hands-on advice for carrying out key recommendations using freely available software and packages in the R environment, including specific functions and packages (see Table 2 for resources on preregistration and other approaches to reducing analytical flexibility, Table 3 for more information about simulation-based power analysis, Table 4 for sample size planning in sequential designs, Table 5 for an overview of effect sizes for popular statistical models, and Table 6 for an overview of biased effect size estimates and recommended alternatives). We also provide a glossary of key terms introduced in the paper at https://osf.io/xtq6s/ and as Supplementary Information accompanying this paper that can be used as a reference while reading this paper or for teaching purposes.

To illustrate how the proposed reporting standards can be applied in practice, we provide a series of best-practice examples embedded throughout the manuscript and summarized in Table 1. These examples, based on a hypothetical cognitive psychology study on visual working memory, contrast common reporting pitfalls with more transparent and

## Box 2 | Different sampling procedures

**Fixed-sample designs:** In a fixed-sample design, the final sample size is predetermined before data collection begins. This idea of a fixed sample size is linked to the frequentist perspective on the concept of statistical power. In this framework, a power analysis is conducted using an effect size of interest to calculate the necessary sample size, ensuring control over the beta error rate in the long run[34]. The data collection process may occur over multiple sessions to obtain the final sample size, but the data analyst is prohibited from stopping data collection before the final data point is collected. This restriction is essential because examining the data before reaching the predetermined sample size—and potentially stopping if a statistically significant result is found—can inflate the Type I error rate when *p*-values are used for inference[230]. This inflation occurs because *p*-values, under the null hypothesis with a continuous and correctly specified null distribution, follow a uniform distribution and because *p*-value inference depends on the stopping rule[231], making such interim analysis in fixed designs a questionable research practice known as optional stopping[230]. Optional stopping is generally considered less problematic in Bayesian statistics, particularly when subjective priors are used, because the inference is conditioned on the data and prior[107,232,233]. However, this view is not universally accepted, and the robustness of results under different stopping rules remains a subject of debate[233]. In general, fixed-sample size designs are most commonly used in psychological research, as they allow for efficient and planned resource allocation during the planning stage. However, these designs can be inefficient in practice. For example, if the true effect size is larger than anticipated, the predetermined sample size is unnecessarily large[118,127].

This inefficiency contrasts with sequential methods, which explicitly plan multiple looks in the data and allow for earlier conclusions.

**Sequential designs:** Sequential designs either use a test statistic that is independent of the sampling plan, such as Bayes factors or likelihood ratios[232,234] or they adjust explicitly for alpha-level inflation[55]. In all approaches, earlier looks into the data are an intentional aspect of the design. Depending on the sequential test, the researcher is permitted to inspect the data at predefined stages in the collection process, such as after reaching a certain number of data points, or at any arbitrary point during data collection. Two well-known sequential test families are Group sequential designs (GSD)[188,189] and sequential likelihood ratio tests such as the Sequential Probability Ratio Test (SPRT)[127,235] and the Sequential Bayes Factor (SBF) [107]. GSDs use *p*-values as test statistic, with interim analyses ("looks") planned in advance and a final sample size known beforehand. More flexible GSDs permit changes of the preplanned looks even during data collection, but only if the decision to take a look is independent of the evidence in the data. In contrast, the SPRT and the SBF have an unknown final sample size but allow looks at any time. Although SPRT and SBF originate from different statistical paradigms, they are conceptually very similar and both highly efficient. Heuristically, a SPRT can require around 50% fewer data points than a fixed-sample test. However, this efficiency gain depends on the specific combination of the expected and true effect sizes: it tends to be smaller when large effects are expected, but can even exceed 50% when small effects are expected[106,118,127]. A sequential design, however, requires greater flexibility in study logistics than a fixed-sample design, as the final sample size is not known in advance.

## Box 3 | Resources for machine learning and qualitative analysis reporting

**Machine learning (ML):** ML is gaining increasing attention in psychological research due to its ability to analyze complex, high-dimensional data and to generate predictive insights that can complement or surpass those derived from traditional statistical approaches[236]. Among the various ML paradigms, supervised learning is particularly relevant for many applications in psychology. This approach involves training algorithms on labeled data to identify patterns and predict outcomes, typically using classification methods for categorical outcomes and regression techniques for continuous ones[236,237]. Although a detailed treatment of machine learning (ML) methodologies is beyond the scope of this paper, several resources are available for readers interested in further exploration. Introductory material on supervised ML tailored to psychological research, along with discussions on study design and sample size considerations, can be found in domain-specific tutorials and methodological reviews[237,238]. Comprehensive overviews of ML reporting standards are also available, including guidelines aimed at improving transparency, replicability, and methodological rigor in ML-based science[239]. In clinical prediction modeling, the TRIPOD+AI statement offers a structured checklist for reporting studies based on both regression and ML techniques[240]. Recommendations for preregistering predictive modeling studies have also been proposed to foster reproducibility and reduce analytic flexibility[241]. For an accessible overview of interpretable machine learning in psychological research, recent work discusses both conceptual foundations and practical tools[242]. As a complementary resource, model—agnostic methods for explaining individual predictions-such as SHAP (SHapley Additive exPlanations) values—are introduced in interpretability-focused contributions to the ML literature[243].

**Qualitative research and mixed methods:** The recent push to improve credibility and reproducibility in psychology has largely been grounded in a quantitative positivist perspective[244,245]. This is evidenced by the majority of credibility and reproducibility tools and practices being designed with quantitative research squarely in mind. While the concepts of preregistering hypotheses or replication are not compatible, or even desirable, for most knowledge production approaches in qualitative research[246], there are several ways to increase transparency in qualitative research[247], which include providing details on how data was produced (e.g., positionality statements, the wider context of data collection[248]) and how data was analysed (e.g., codebooks[249,250]). Because reporting guidelines developed for quantitative research are often ill-suited to qualitative studies, dedicated checklists have been introduced to support transparent reporting of both primary qualitative research and qualitative evidence syntheses[251–254]. Broader discussions of transparency from a qualitative standpoint are available in recent methodological contributions[248,250].

**Table 1 | Insufficient vs. best-practice statistical reporting in a hypothetical cognitive psychology study (visual working memory)**

| Reporting Element | Insufficient Reporting Example | Best-Practice Example |
|---|---|---|
| Hypothesis and Design | "We expected that the high-load condition would affect working memory." | "We hypothesized that participants in a high-load visual working memory condition would have lower recall accuracy (and longer RTs) than those in a low-load condition." |
| Preregistration | "The methods were preregistered online." | "We preregistered all hypotheses and planned analyses on the Open Science Framework prior to data collection (on 05/11/2025); any deviations from that preregistered plan are noted in the Results." |
| Sample Size Justification | "We recruited 40 undergraduates." | "Our a priori power analysis (80% power, $\alpha = 0.05$) for detecting a medium effect size ($d = 0.50$) required 52 participants. We oversampled to 60 in anticipation of attrition." |
| Outlier and Missing Data | "We excluded extreme results and participants who failed to finish." | "In line with our preregistered plan, reaction times exceeding 3 SDs from the condition mean were treated as outliers and removed from subsequent analyses, affecting 2.5% of trials. Five participants who withdrew mid-study were excluded from final analyses (final $N = 55$)." |
| Statistical Model Specification | "We conducted an ANOVA and found a difference between experimental conditions." | "A $2 \times 2$ repeated-measures ANOVA (Condition: high-load vs. low-load; Time: pre vs. post) on recall accuracy was conducted. The assumption of normality was examined and met." |
| Software and Code Disclosure | "We used R to analyze the data." | "All analyses were performed in R (v4.2.1) using the `afex` package (v1.1). The analysis code and de-identified data are openly available (e.g., on an OSF repository) to facilitate reproducibility." |
| Inferential Statistics | "We observed a significant effect on recall accuracy ($p < 0.05$)." | "A significant Condition × Time interaction emerged for recall accuracy, $F(1, 54) = 4.37$, $p = 0.04$, partial $\omega^2 = 0.06$ (95% CI [0.00, 0.23]). Post hoc analyses revealed that accuracy decreased significantly from pre- to post-test in the high-load condition ($M$ difference $= -0.15$, $SE = 0.04$, $p < 0.001$, $d = 0.68$ (95% CI [0.39, 0.97])), while accuracy did not significantly differ in the low-load condition ($M$ difference $= -0.02$, $SE = 0.04$, $p = 0.630$, $d = 0.09$ (95% CI [$-0.18$, 0.36])). This pattern indicates that memory load impaired performance over time, with a medium-to-large effect size for the decline in the high-load condition. |
| Null Results | "We observed no statistically significant effect of load on RT ($p = 0.16$). This indicates that memory load had no effect on participants' RT." | "We observed no significant effect of load on RT, $F(1, 87) = 0.01$, $p = 0.92$, partial $\omega^2 = 0.00$ (95% CI [0.00, 0.01]). We further ran an equivalence test (TOST) using $\pm 0.20$ as our smallest effect size of interest, and the 90% CI for $d$ was fully contained within these bounds, suggesting the effect of load on RT is practically negligible. Both the lower-bound test, $t(87) = 2.08$, one-sided $p = 0.020$, and the upper-bound test, $t(87) = -1.67$, one-sided $p = 0.049$, were statistically significant." |

rigorous alternatives. By integrating these concrete examples at key points-such as hypothesis formulation, preregistration, outlier handling, and the interpretation of null results-we aim to bridge the gap between abstract recommendations and practical implementation.

To further support implementation, we provide the Transparent Statistical Reporting in Psychology (TSRP) Checklist as a freely available resource at https://osf.io/t2zpq/ and as Supplementary Information accompanying this paper. This checklist aligns with the paper's recommendations and offers a structured format for planning, documenting, and reviewing statistical analyses. While consensus-based tools like the Transparency Checklist offer broad open science coverage[4], the TSRP checklist focuses specifically on the technical and inferential aspects of statistical reporting in quantitative psychology. Together, these resources contribute to more transparent and interpretable research practices.

## Reporting of study design and analysis plans

In this section, we outline the key decisions that one typically faces before data collection and analysis. These considerations include defining the hypotheses and research questions, planning sample sizes, addressing outliers and missing data, selecting statistical models and tests, and choosing software for conducting analyses.

### Confirmatory vs. non-confirmatory research

When planning a study design and later reporting inferential statistics, one should always distinguish between confirmatory hypothesis testing and non-confirmatory (exploratory) research. Even within purely confirmatory research designs, researchers may conduct additional exploratory analyses, such as assessing the influence of moderators or evaluating different pre-processing choices on a main statistical finding. Such post hoc analyses

should be explicitly labeled as non-confirmatory. Conversely, non-confirmatory research papers can be either purely discovery-oriented or aimed at addressing foundational aspects that strengthen a derivation chain for hypothesis testing[7,8]. Such foundational work might include developing and validating psychological measures or experimental manipulations[9], examining preprocessing and analysis pipelines[10], or synthesizing research to establish robust benchmark findings for the field[11], among other possible topics.

Many of the reporting principles outlined here, such as transparent documentation of preprocessing decisions, effect size reporting, and clear visualization of results, are equally beneficial for confirmatory and non-confirmatory studies. Transparent reporting in non-confirmatory work helps readers assess the reliability of the findings and facilitates their translation into future confirmatory research.

### Formulating testable hypotheses

Precise and operationalizable hypotheses are essential for theory-testing confirmatory research. However, psychological theories are often ambiguous, which hinders cumulative knowledge building[12]. A major issue is the reliance on imprecise, ad hoc verbal hypotheses. These often lack specificity regarding the variables involved, the expected direction or magnitude of effects, and the underlying mechanisms, thereby allowing excessive interpretative flexibility.

For instance, a researcher might hypothesize that "stress impairs memory performance". While seemingly intuitive, this claim is underspecified: it does not define what kind of stress is being studied (e.g., acute vs. chronic, physiological vs. perceived), how memory is assessed (e.g., recall vs. recognition, short- vs. long-term), or the expected size of the effect. Moreover, without specifying a mechanism (e.g., cortisol-induced disruption of

**Table 2 | Resources for preregistrations and other approaches to reducing analytical flexibility**

| | |
|---|---|
| Preregistration templates | AsPredicted Template: AsPredicted.org. (n.d.). AsPredicted: A standardized format for preregistration. https://aspredicted.org[170]. |
| | OSF Preregistration Template: Mellor, D. T., Esposito, J., Hardwicke, T. E., Nosek, B. A., Cohoon, J., Soderberg, C. K., Kidwell, M. C., Clyburne-Sherin, A., Buck, S., & DeHaven, A. C. (2015). OSF-Prereg-template.docx. https://osf.io/jea94[171]. |
| | PRP-QUANT Template: Preregistration Task Force (2024). Preregistration standards for psychology-the Psychological Research Preregistration-Quantitative (aka PRP-QUANT) Template. https://www.psycharchives.org/en/item/a417b468-7398-40e2-9050-4adad9e2078d[172]. |
| Reporting deviations | Lakens (2024)[25] |
| | Willroth & Atherton (2024)[26] |
| Decision trees | Nosek et al. (2018)[20] |
| Blind analysis | MacCoun & Perlmutter (2015)[27] |
| | Dutilh et al. (2021)[173] |
| Data splits | Vermeent et al. (2024)[28] |
| Multiverse analysis | Harder (2020)[30] |
| | Steegen et al. (2016)[29] |

hippocampal function), almost any outcome can be interpreted as consistent with the hypothesis. If memory improves, stress might be said to enhance focus; if it declines, it might be blamed on cognitive overload. This kind of ambiguity impedes falsifiability and enables post hoc rationalization, ultimately undermining theoretical progress[13]. These concerns have prompted growing calls for greater rigor in hypothesis formulation[12,14].

One strategy to enhance theoretical clarity is formalization, where verbal hypotheses are translated into mathematical models or computational frameworks[15,16]. Formalized hypotheses explicitly specify variables, relationships, and expected outcomes, reducing ambiguity and ensuring internal consistency[12,17]. For instance, rather than making a vague directional prediction such as "higher cognitive load impairs performance," a formalized hypothesis might predict a specific numerical relationship between cognitive load and reaction time. More elaborate models may include equations that map proposed mechanisms—such as how memory load influences decision-making speed and accuracy—onto specific mathematical parameters. This approach limits researcher degrees of freedom and enhances opportunities for falsifiability.

Greater theoretical precision also starts with conceptual clarity (i.e., using well-defined constructs and careful language when formulating hypotheses). Recent work highlights that many psychological concepts are poorly defined, leading to ambiguous hypotheses and inconsistent measurement practices[18]. Researchers should explicitly link theoretical constructs to their operationalizations, explaining why a specific measure, such as a survey score or behavioral indicator, captures the intended construct. If a construct can be measured in multiple ways, researchers should acknowledge these variations and justify their chosen approach. For example, instead of the vague statement, "*Working memory capacity was assessed using a digit span task,*" a more transparent description would be: "*Working memory capacity was assessed using a backward digit span task. While several tasks could be used to assess this construct (e.g., complex span, n-back), we selected this task due to its widespread use and strong psychometric properties in prior visual-verbal memory research.*"

Beyond general calls for clarity, several recent publications propose structured frameworks to guide researchers in formulating hypotheses more rigorously[15,19]. One such approach emphasizes the scope of a hypothesis, which can be defined by the nature of the relationship being tested, the specific variables involved, and the data processing or analysis pipeline[19]. A well-formulated hypothesis should balance specificity and generalizability, making clear, testable predictions while remaining theoretically meaningful. For example, a study may examine visual working memory under different load conditions. A vague hypothesis such as "*We expected that the high-load condition would affect working memory*" leaves the direction and nature of the effect unspecified, allowing for excessive interpretative flexibility. In contrast, a well-formulated hypothesis might state: "*We hypothesized that participants in a high-load visual working memory condition would have*

*lower recall accuracy (and longer RTs) than those in a low-load condition.*" This version operationalizes both the predictor and outcome, specifies the expected direction of the effect, and allows for clear falsifiability.

Hypotheses that are too narrow risk becoming trivial, whereas those that are too broad may be unfalsifiable. For instance, "*We hypothesize that cognitive control influences behavior*" is overly broad and unfalsifiable, whereas "*We expect a 35-ms difference in RT between conditions on trial 3*" is overly narrow and likely trivial. A more balanced hypothesis might be: "*We hypothesize that participants with higher working memory capacity will show reduced switch costs (as indexed by reaction time differences) in a task-switching paradigm compared to participants with lower capacity.*" Researchers should strive for precision while ensuring their hypotheses remain relevant and applicable to the broader theoretical context.

### Preregistration

Preregistration involves specifying key study decisions *before* data collection or analysis[20]. When feasible, preregistering hypotheses, study design, and analysis plans can help reduce bias, limits researcher degrees of freedom, and enhances the credibility of findings[10]. Within Mayo's *error statistical* framework, which focuses on how statistical methods can expose potential errors in hypothesis testing, preregistration allows for more *severe* hypothesis testing: the more specific and falsifiable a preregistered claim (e.g., a predicted effect size range), the stronger its inferential value[21]. A time-stamped, pre-specified analysis plan further constrains researcher flexibility, reducing opportunities for post hoc adjustments that could inflate false positives. Importantly, preregistration can still be useful even when data have already been collected, such as when working with pre-existing datasets[22]. In addition, preregistration can also be a useful tool in exploratory work, where researchers may preregister the planned preprocessing and data analysis steps even if they plan no confirmatory hypothesis tests[23]. Table 2 contains many useful resources for preregistrations referred to in the text. In addition, the Center for Open Science (COS) provides many further helpful resources at https://www.cos.io/initiatives/prereg.

Post hoc analyses, including deviations from preregistered plans or exploratory analyses, are often necessary—for instance, when statistical models fail to converge or distributional assumptions are violated[24]. Such deviations are normal and expected, as oversights in the planning phase or analytical challenges are often only revealed during data collection or analysis. Transparent documentation of such deviations allows readers to assess their impact, which can promote credibility and methodological rigor[25,26]. For preregistration statements to serve this purpose, they must include sufficient detail. A vague note such as "*The methods were preregistered online*" provides little insight into what aspects were registered and where. In contrast, best practice would involve a statement like: "*We preregistered all hypotheses and planned analyses on the Open Science Framework prior to data collection; any deviations from that preregistered plan are noted in the*

*Methods and Results*." This level of detail supports traceability and ensures readers can verify adherence to the preregistered plan.

Preregistration may be challenging in some cases, for example, in data-driven research or studies involving complex statistical models. In these situations, researchers can preregister aspects of the analysis plan that can be determined in advance, such as data preprocessing steps or outlier exclusion criteria. Alternatively, a decision tree can be preregistered to outline decision rules for each stage of a data-driven analysis procedure[20]. Another approach involves blinding the dataset by scrambling some observations, preserving distributional properties while concealing actual outcomes. This allows researchers to address outliers and modeling assumptions without revealing results until the dataset is unblinded[27]. Lastly, researchers can also split pre-existing datasets into two parts, using the first part to assess distributional properties and modeling assumptions and then incrementally preregistering their analysis plans for the second part[28].

While preregistration promotes transparency and reduces hindsight bias, it does not automatically improve research quality. Registering one analysis plan from multiple valid alternatives can be arbitrary and does not necessarily strengthen inferential validity[12]. Preregistration cannot by itself correct flawed analytical choices, highlighting the importance of thoughtful design and justification alongside preregistration. In cases where multiple reasonable analysis paths exist, a multiverse analysis—examining the robustness of findings across different preprocessing and statistical approaches—may provide more informative insights[29,30]. Ultimately, while preregistration can serve as a valuable mechanism for enhancing transparency and rigor within an 'error statistical' framework[21], it should not be regarded as universally necessary, particularly when it risks favoring a single arbitrary perspective over a more comprehensive examination of the data[31].

## Sample size planning

Researchers must clearly justify their study's sample size and composition when designing and reporting a study. For fixed-sample designs conducted within a frequentist framework, researchers must decide on the sample size before data collection, based on considerations such as statistical power and estimation precision. In contrast, sequential tests provide more flexibility but still require researchers to determine a sequential stopping rule.

In this section, we outline the key aspects of sample size planning that researchers should consider when designing a study. We also emphasize the importance of transparency in reporting the rationale for the chosen sample size. This rationale should be clearly documented, either by describing the procedure in the text or by sharing an analysis script that allows others to reproduce the underlying calculations.

In cases where data have already been collected, sensitivity analysis offers a meaningful alternative to traditional power analysis. Rather than estimating the required sample size, sensitivity analysis identifies the smallest effect size that the study is capable of reliably detecting given the available sample, chosen alpha level, and desired power[32]. This approach is particularly useful for contextualizing null results or assessing the robustness of observed effects. Notably, this is preferable to post hoc power analysis, which is widely discouraged due to its misleading interpretation of non-significant results[33] (see Section "Common errors and misinterpretations" below).

For prospective studies, researchers should report how they determined their sample size. Let us consider two approaches to reporting: One might simply state, "*We recruited 40 undergraduates,*" which omits any justification for that number. A better practice is to explain the rationale, as in: "*Our a priori power analysis (80% power, $\alpha = 0.05$) for detecting a medium effect size ($d = 0.50$) required 52 participants. We oversampled to 60 in anticipation of attrition.*" This makes the assumptions behind the sample size transparent and allows others to evaluate the study's statistical validity.

**Frequentist statistics: Power analysis and pitfalls.** When researchers plan to analyze their data using the Neyman–Pearson frequentist framework, they must first determine their desired statistical power. In this framework, statistical power is the long-run probability of rejecting the null hypothesis when a specific alternative hypothesis is true, and it is given by $1 - \beta$[34]. As this is a decision-making theory, rejection of the null is interpreted as acting as though the alternative hypothesis is true, without asserting its truth. Statistical power depends on the statistical test, the planned sample size, the effect size specified under the alternative hypothesis, and the $\alpha$ level. An a priori power analysis is therefore conducted before data collection to determine the sample size required to achieve the desired power. If fewer data are collected than planned, the power to detect the specified effect is reduced. If substantially more data are collected, the power increases, making it more likely to detect effects smaller than those originally targeted, which may not be of substantive interest. Consequently, the planned sample size is a critical element of the Neyman–Pearson testing procedure and should align with the study's theoretical and practical goals.

When determining the required sample size, researchers have to specify the specific statistical test they intend to use (e.g., a bivariate correlation, a within-subjects $t$-test, or a 3-way ANOVA), the statistical power they aim to achieve (usually at least $1 - \beta = 0.80$, but a higher power can be desirable if researchers want to be more confident about their findings—for example in replication studies), the desired significance level (usually set to $\alpha = 0.05$), and the effect size they anticipate to observe (see next section for a discussion of where expected effect sizes can be obtained and common pitfalls in using them). There are three aspects that can complicate this seemingly straightforward procedure of sample size planning: Limited or biased knowledge about the expected effect size, multiple statistical tests in a single study, and more complex statistical modeling approaches.

*Limited or biased knowledge about the expected effect size.* The prior knowledge about an expected effect size is sometimes limited (if there is only little previous research one is building upon) and often biased due to a combination of questionable research practices (QRPs), small-sample designs and publication bias in the published literature. These biases can lead to an overestimation of effect sizes, which, in turn, skews the planning of new studies.

QRPs contribute to the inflation of effect sizes in the literature[10]. Researchers may unintentionally (or sometimes deliberately) manipulate their data analysis procedures to achieve statistically significant results. For example, repeatedly running analyses until a significant $p$-value is obtained or selectively reporting only those outcomes that support a desired hypothesis can result in effect sizes that do not reflect the true magnitude of the phenomenon under study[35]. These inflated effect sizes, when used for future research, can lead to underpowered study designs and test combinations that are less likely to replicate.

Small-sample designs further compound the problem. Studies with small sample sizes are more prone to sampling error, which can lead to large variability in effect size estimates. As a result, statistically significant findings are more likely to overestimate the true effect, particularly when the true effect is small or modest[36]. Additionally, small-sample studies are often statistically underpowered to detect effect sizes of interest, which not only reduces the likelihood of detecting true effects but also contributes to the 'file drawer problem', where null results that often correspond to underestimates of the true effect due to sampling variability are not published, further distorting the literature.

Publication bias, where studies with significant findings are more likely to be published than those with null or negative results, exacerbates these issues. As a result, the published literature in psychology tends to present a biased view[37], which is associated with effect size inflation[38–40]. This bias can mislead researchers when planning new studies, leading them to expect larger effects than are warranted, and thus to plan insufficiently powered studies.

Given these challenges, researchers should be cautious in using previously reported effect sizes for sample size planning[32]. It is crucial to critically evaluate the literature for evidence of publication bias and consider the context in which effect sizes were obtained. To ensure that test and study design combinations are not statistically underpowered, researchers can

**Table 3 | Further readings and software recommendations for simulation-based power analysis across common statistical models**

| Statistical model family | Further reading | R packages |
|---|---|---|
| Analysis of variance | Brysbaert (2019)[174] Caldwell et al. (2022)[175] | `simr`[176], `Superpower`[177] |
| Structural equation models | Irmer et al. (2024)[178] Moshagen & Bader (2023)[179] | `semTools`[180], `lavaan`[181], `simsem`[182], `semPower`[179] |
| Multilevel models | Green & MacLeod (2016)[176] | `simr`[176], `lme4`[183] |
| Network models | Constantin et al. (2023)[184] | `bootnet`[185], `psychonetrics`[186], `EGAnet`[187] |

conduct a safeguard power analysis, which uses a lower-bound estimate of the effect size based on the estimate's uncertainty for sample size planning[41]. In addition, consulting a meta-analysis may provide a more reliable estimate of the effect size than relying on a single estimate. However, summary effect size estimates from meta-analyses can also be inflated due to publication bias[42], which in turn influences sample size planning, unless a correction for publication bias is used, such as Robust Bayesian meta-analysis[43] or selection models[44].

In the absence of prior research, one common but problematic approach is to estimate effect sizes from a pilot study. Because pilot studies typically involve small samples, they tend to produce highly unstable and biased effect size estimates, leading to inaccurate sample size planning[45]. Unless a pilot study is conducted with a sufficiently large sample—something often impractical—it provides little reliable information about expected effect sizes.

An increasingly recommended alternative is to define the minimal effect size of interest based on theoretical or practical considerations[46]. For example, instead of stating, "*We based our sample size on an expected medium effect size,*" a more informative approach is: "*We defined a minimal effect size of interest as d = 0.30, based on prior meta-analytic estimates and the smallest effect that would have theoretical and applied relevance for the cognitive training intervention.*" Rather than relying on uncertain empirical estimates, this approach anchors sample size decisions to meaningful effect thresholds that reflect substantive or applied significance. This strategy is particularly useful when prior research is sparse and conducting a large pilot study is infeasible, ensuring that studies are designed to detect effects that are genuinely meaningful rather than statistically arbitrary.

Multiple statistical tests. Power analysis becomes more challenging when conducting multiple statistical tests (e.g., correlations and ANOVAs), as researchers must decide which analysis will guide their sample size planning. Generally, it is advisable to base the sample size on the test that requires the largest sample to ensure adequate power across all analyses. For example, rather than stating, "*Sample size was determined based on the main ANOVA*", one could write: "*Sample size was based on the mediation analysis, which required a larger sample than the primary group comparison, ensuring sufficient power across all planned analyses.*" Tests should only be deprioritized in sample size planning if they are intended for purely exploratory purposes.

When multiple statistical tests are planned, multiplicity adjustments (such as Bonferroni or Holm corrections) can be pre-specified during the design stage to control the family-wise Type I error rate. Incorporating these adjustments into sample size planning ensures that the study remains adequately powered for each test while accounting for the more stringent significance thresholds required to address multiplicity.

More complex statistical modeling approaches. While sample size planning is technically straightforward for simple statistical modeling approaches (e.g., t-tests, one-factorial ANOVAs, correlations, etc.), power analysis becomes quickly more complicated for more complex modeling approaches such as multiway ANOVAs, structural equation models (SEMs), multilevel model (MLMs), network analyses, and many other models. These complex designs often involve multiple parameters and hierarchical structures, making traditional power analysis methods insufficient or impractical. In

such cases, Monte Carlo simulation offers a powerful and flexible approach to estimate power and determine appropriate sample sizes.

Monte Carlo simulations involve generating data that closely mimics the real-world data structure expected under a hypothesized model. By specifying the model parameters, including effect sizes, variances, correlations, and the number of observations, researchers can simulate numerous datasets that reflect different possible outcomes under the model. These simulated datasets are then analyzed using the intended statistical methods to assess how often the correct conclusions (e.g., detecting a true effect) are reached. This process can be repeated thousands of times to estimate the statistical power of the analysis for a given sample size (which is calculated as the percentage of significant findings across repetitions) and can be repeated for different sample sizes to determine the optimal sample size to achieve the desired statistical power for all or the most relevant parameters of a model (e.g., $1 - \beta = 0.95$). Table 3 provides further information and recommendations regarding simulation-based power analysis for several classes of popular statistical models.

**Bayesian frameworks: precision and simulation.** Traditionally, power analysis is not considered essential in Bayesian data analysis. This is because Bayesian inference focuses on updating beliefs, with any additional data contributing valuable information to this process[47]. However, researchers using a Bayesian framework still need to determine the amount of data they plan to collect, unless they are employing a sequential design (discussed below). Unlike frequentist approaches, Bayesian power analysis can incorporate various decision criteria, such as ensuring a desired level of precision in estimating an effect size or simply testing for the existence of an effect[48].

Bayesian sample size determination can be performed using either simulation-based or analytical approaches, depending on the complexity of the model and the planning criterion. For simpler or conjugate models, analytical methods can determine the required sample size to achieve a target posterior precision or a pre-specified Bayes factor without extensive simulations[49–51]. In more complex or non-conjugate settings, simulation remains a flexible approach.

There are several approaches to generating data for power analyses in Bayesian frameworks[48,52]. One method involves creating a distribution of effect sizes, such as Cohen's d, which reflects an anticipated range of effect sizes. This could represent a posterior distribution from prior research or an informed estimate based on the field's knowledge. Alternatively, data can be generated by running a Bayesian analysis on previously collected data and then using the posterior distribution from that analysis to create new data. Another approach is to generate data based on a range of specific effect sizes of interest.

Subsequently, the statistical model is applied to the simulated datasets, and the probability of achieving a specific inference goal is calculated (e.g., determining whether the highest density interval excludes the smallest effect size of interest or assessing the width of a credible interval for a particular parameter). Moreover, researchers can examine the rate of misleading evidence (cases where the data strongly favor the wrong hypothesis) depending on their simulation parameters[53]. Additionally, they may want to conduct a *prior sensitivity analysis* to evaluate how different prior distributions influence the inference goal across various sample sizes[54].

**Table 4 | Comparison of sequential procedures with respect to sample size planning**

| Category | Group Sequential Design (GSD) | Sequential Probability Ratio Test (SPRT) | Sequential Bayes Factor Test (SBFT) |
|---|---|---|---|
| Framework | Frequentist | Frequentist | Bayesian |
| Error control | Type I & Type II | Type I & Type II | No formal long-term error control |
| Sequential steps | The steps need to be fixed in advance when classical approaches are used[188,189]. More flexible methods are available for more complex scenarios (e.g., alpha-spending functions[55]), though these approaches are still not fully sequential. | Fully sequential; looks do not need to be planned in advance | Fully sequential; looks do not need to be planned in advance |
| Final sample size | Known beforehand | Not known beforehand | Not known beforehand |
| Average sample number (ASN) efficiency | On average more efficient than fixed designs but less efficient than fully sequential designs | Very efficient | Very efficient |
| Planning the sample size | Based on the power analysis result of a fixed design multiplied with the inflation factor of the specific design[55], or by using software mentioned below. | One way is to perform a power analysis for a fixed-design to plan the maximal sample size, as SPRTs usually reach a decision in 90% of time before the fixed sample size is reached[118]. Alternatively, simulations can be performed to get insights about the ASN . | Usually based on ASN simulations[190]. |
| R packages | *Rpackt*[131], *gsDesign*[191] | *sprtt*[134] | *BayesFactor*[94], *BFDA*[192] |

Overall, Bayesian sample size planning offers a flexible framework that enables researchers to tailor their analyses to specific research objectives, such as achieving desired precision in parameter estimates or incorporating prior knowledge. While this approach has clear advantages, it also demands careful attention to prior distributions, computational complexity, and modeling choices. For researchers new to Bayesian inference, several tutorials provide valuable guidance on Bayesian power analysis and prior sensitivity analysis, making them an excellent starting point[48,52–54].

**Sequential designs: Maximal sample sizes and average sample number.** Sequential designs offer a flexible and efficient alternative to fixed-sample approaches by permitting interim analyses of accumulating data. These designs allow researchers to reach conclusions earlier-either by stopping for efficacy when sufficient evidence has been obtained or for futility when meaningful effects are unlikely. Popular sequential methods include Group Sequential Designs (GSDs), Sequential Probability Ratio Tests (SPRTs), and Sequential Bayes Factors (SBFs), each of which differs in its statistical framework, stopping logic, and sample size implications.

Table 4 summarizes the key characteristics of these designs, including whether the final sample size is known in advance, their relative efficiency, and how the required sample size is typically planned. In general, GSDs retain a maximum sample size and pre-specified number of interim looks, with control over the Type I error rate via alpha-spending functions (which control the cumulative Type I error rate across interim analyses) and over the Type II error rate with optional beta-spending functions (which manage the Type II error rate)[55]. In contrast, SPRTs and SBFs allow fully flexible stopping but do not pre-define a final sample size. Instead, stopping occurs when likelihood ratios or Bayes factors cross specified decision thresholds. However, recent simulation studies of Bayesian sequential testing have highlighted potential issues associated with interim analyses-particularly when vague or noninformative priors are used[56]. One potential remedy is the use of more conservative priors, which can mitigate some of these problems.

Careful planning is essential for all sequential methods. For GSDs, maximum sample sizes can be determined by applying an inflation factor to a conventional power analysis. For SPRTs and SBFs, where no maximum sample size is specified, simulation-based planning is strongly recommended to estimate the average sample number (ASN) for different effect size scenarios. The ASN refers to the expected number of observations required to reach a decision under a given design and effect size. It is often used to compare the efficiency of sequential versus fixed-sample designs.

Table 4 provides further technical guidance for implementing these simulations using freely available R packages.

**Measures and experimental design**
Measurement can be defined as "any approach that researchers take to create a number to represent a variable under study"[9]. Measuring a construct that is meant to be measured is integral to ensure the validity of the conclusion of a study. However, defining a construct in psychological research can be challenging because it is often not directly observable[9], and there are diverse ways to measure the construct[57,58]. Despite its importance, the validity of measurement tools remains frequently underreported across disciplines[57,59,60]. Common issues include omitting references to the instruments used, failing to provide evidence that the tools measure the intended constructs, and making undocumented modifications to existing measures. These omissions undermine the ability to evaluate a study's psychometric soundness and, ultimately, weaken the strength of its empirical claims[61].

All measures—primary, secondary, and covariates—should be reported, even if they are not included in the final analysis[3]. Comprehensive reporting helps guard against selective reporting and enhances transparency. In addition, psychometric properties should be documented, including reliability (e.g., interrater reliability, test-retest correlation, split-half correlation, internal consistency) and evidence of validity (e.g., references to validation studies, justification and documentation of modifications to established measures)[9]. A checklist to support the validity of individual studies is provided by Kerschbaumer et al. (2025)[62].

Likewise, if a study involves experimental manipulations, the details of these manipulations should be reported[3]. This information entails the content of the experimental manipulation and the method, time and location of manipulation delivery, including a description of all experimental and control conditions. Detailed information on measurement and experimental manipulations is critical to interpreting the results and replicating a study[63]. It is also essential to include a manipulation check—an empirical test to verify that an experimental manipulation successfully influenced the intended construct—to ensure that the manipulation targets the construct as intended[64].

**Handling outliers and missing data**
Outliers refer to data points that deviate substantially from the other observations in the dataset[65] and can affect the interpretation of relationships between variables if not addressed[66]. However, there is no consensus about defining outliers and detecting them[67]. If researchers define and

handle outliers in the data set depending on observed outcomes, it can substantially bias the analysis, leading to an inflated Type I error rate[65,66]. Therefore, both the criteria for detecting outliers and the methods to address outliers should be defined in advance of data collection and reported[66]. Authors should also provide the frequency or percentage of data excluded from analysis and detailed reasons (e.g., participants and/or data points, dropouts)[3].

In some cases, however, it may not be necessary—or even appropriate—to remove outliers from the dataset. Rather than excluding data points based solely on deviation from central tendencies, researchers may consider applying robust statistical methods that reduce the undue influence of extreme values while preserving the integrity of the full dataset[68]. Robust estimators, such as M-estimators or trimmed means, which reduce the influence of extreme values on parameter estimates, and nonparametric resampling techniques such as bootstrapping, a resampling method that repeatedly draws samples from the observed data to estimate the sampling distribution, offer alternatives that can accommodate outliers without violating model assumptions or inflating Type I error rates[69]. These approaches are particularly useful when outliers reflect valid individual differences or meaningful experimental variation rather than measurement error. As with exclusion criteria, the use of robust or resampling methods should be justified and pre-specified to ensure transparency and reproducibility in data analysis.

Missing data refers to values that are unobserved but otherwise would be of interest[70]. It is a common problem in psychological research, and addressing missing data is important for ensuring the precision and validity of research findings. For transparent reporting, criteria for deciding when to infer missing data and how these are handled in data analyses should be decided in advance to data collection and reported[3].

Missingness of data can be explained in different mechanisms: missing completely at random (MCAR), missing at random (MAR), and missing not at random (MNAR)[71]. Each type presents different statistical challenges. When data is MCAR, the reason behind missing observation is unrelated to observed data or missing values[71]. This may reduce the precision of results due to reduced sample size but generally does not lead to biased results[71]. In contrast, in the case of MAR (i.e., when missing data is dependent on the observed data) or MNAR (i.e., when missing data is related to the unobserved data), it can affect the validity of analysis (e.g., due to imbalanced baseline differences[72]) and/or introduce biases[70]. Therefore, understanding the reasons behind missing data and planning how to handle it in advance of data collection is important. Methods for handling missing data should be guided by the type of missing data[70].

Complete case analysis (also known as list-wise deletion) is the most commonly reported missing data treatment method[73,74]. Though this method can be useful in case of MCAR, this method has been advised against in other cases due to substantial loss of data and generalizability of the results[75]. Instead, multiple imputation[71,76], a method that replaces missing values with a set of plausible values based on observed data, or full information maximum likelihood[77], which estimates model parameters directly using all available data without imputing missing values, are recommended to address missing data[74,78] (see Lee and Shi (2021)[79] for a comparison of the two methods).

Transparent reporting of data exclusions and outlier treatment is essential. For instance, the vague statement "*We excluded extreme results and participants who failed to finish*" leaves readers without criteria or justification. A best-practice example might state: "*In line with our pre-registered plan, reaction times exceeding 3 SDs from the condition mean were treated as outliers and removed from subsequent analyses, affecting 2.5% of trials. Five participants who withdrew mid-study were excluded from final analyses (final N = 55).*" This provides clarity about both criteria and impact, supporting reproducibility and critical evaluation.

### Statistical model specification
When planning their sample size and conceptualizing their study, researchers must decide which statistical models and tests best address their research questions. This choice depends on the type of data collected, the assumptions of different statistical methods, and whether the study follows a frequentist, Bayesian, or sequential approach (or a combination thereof). This section discusses best practices for specifying and reporting statistical models.

The statistical models and tests must be described in sufficient detail to allow reproduction of the analyses. Researchers should specify the statistical software, packages, and versions used, and include the analysis code as supplementary material or in a public repository when possible. When analyses are conducted using graphical user interfaces (e.g., SPSS or JASP), the corresponding syntax or analysis file should be generated and shared to enhance reproducibility. The `grateful` R package can be used for generating citation reports, which automatically compile references for the R packages used in an analysis[80]. Even more simply, citations can be obtained using `citation(package = "packagename")` in base R.

Criteria for statistical inferences, such as cut-off values for *p*-values or Bayes factors, should be explicitly stated. Researchers should assess whether the assumptions of the chosen statistical models are met using appropriate diagnostic tests or visualizations, and report any violations. If assumptions are violated, researchers should describe how these were addressed—for example, by retaining the model if it is known to be robust, by using a nonparametric alternative, or by applying robust estimation methods such as bootstrapping when violations are likely to bias the results. In addition, researchers should document how missing data were handled and whether corrections for multiple comparisons were applied. For complex models, one should describe how model fit was evaluated, such as through fit indices, predictive validity with a held-out sample, or model comparisons. Generic descriptions like "*We conducted an ANOVA and found a difference between experimental conditions*" fall short of best reporting standards. A more informative description would be: "*A 2 × 2 repeated-measures ANOVA (Condition: high-load vs. low-load; Time: pre vs. post) on recall accuracy was conducted. The assumption of normality was examined and met.*" This level of specificity allows for replication and helps readers assess the appropriateness of the model used.

## Reporting of results
In this section, we give recommendations for how to report findings. We address the most effective ways to visualize descriptive results, how to report statistical results for frequentist and Bayesian analyses, sequential designs, and meta-analyses, how to calculate, report, and interpret effect sizes, and how to deal with null findings. Even when analyses are exploratory and do not involve formal hypothesis testing, reporting descriptive statistics, effect sizes, and analytical decisions transparently enhances the interpretability and cumulative value of the research. Labeling analyses as non-confirmatory or exploratory signals to readers that findings are hypothesis-generating rather than confirmatory.

### Reporting descriptive statistics
Descriptive statistics summarize the study sample and measures in numbers or figures, or both[81]. Detailed descriptions of the data help reviewers and other readers understand the study findings. Furthermore, it can help meta-analysts calculate effect sizes in cases the effect sizes of interest are not already reported in the study. Therefore, measures of descriptive statistics such as central tendency (e.g., mean, median, or mode) and dispersion (e.g., standard deviation) of the data should be reported for all variables included in the analyses[3].

Visualization of descriptive statistics complements understanding and interpreting the data characteristics. However, each data visualization method has its own strengths and weaknesses, which requires researchers to consider data characteristics carefully[82]. Conventionally, barplots have been used to show central tendency. However, barplots make it difficult for readers to see the data distribution[83,84]. Alternative methods, such as the boxplot[84], raincloud plot[83], and violin plot[85] can be more informative to examine data distributions, as illustrated in Fig. 1.

Reproducible plots that effectively summarize datasets can be generated using R packages such as ggplot2[86]. The R Graph Gallery (https://r-

**Fig. 1 | Impact of visualization choice on data interpretation: Four visualization approaches for group comparisons.** A dataset was generated showing simulated data points in two groups A and B. Group A (shown on the left side of all panels in green) follows a normal distribution with a mean of 10 and a standard deviation of 1, while Group B (shown on the right side of all panels in yellow) exhibits a bimodal distribution, characterized by peaks at 9 and 11 with a standard deviation of 0.5. As shown in the figure (**A**), barplots often fail to capture the true distribution of the data. Although boxplots in (**B**) provide more insights into the data distribution compared to barplots, violin plots in (**C**) and raincloud plots in (**D**) provide a more accurate representation of the actual data distribution. Reproducible code to generate this figure is available at https://osf.io/mk5qa/.

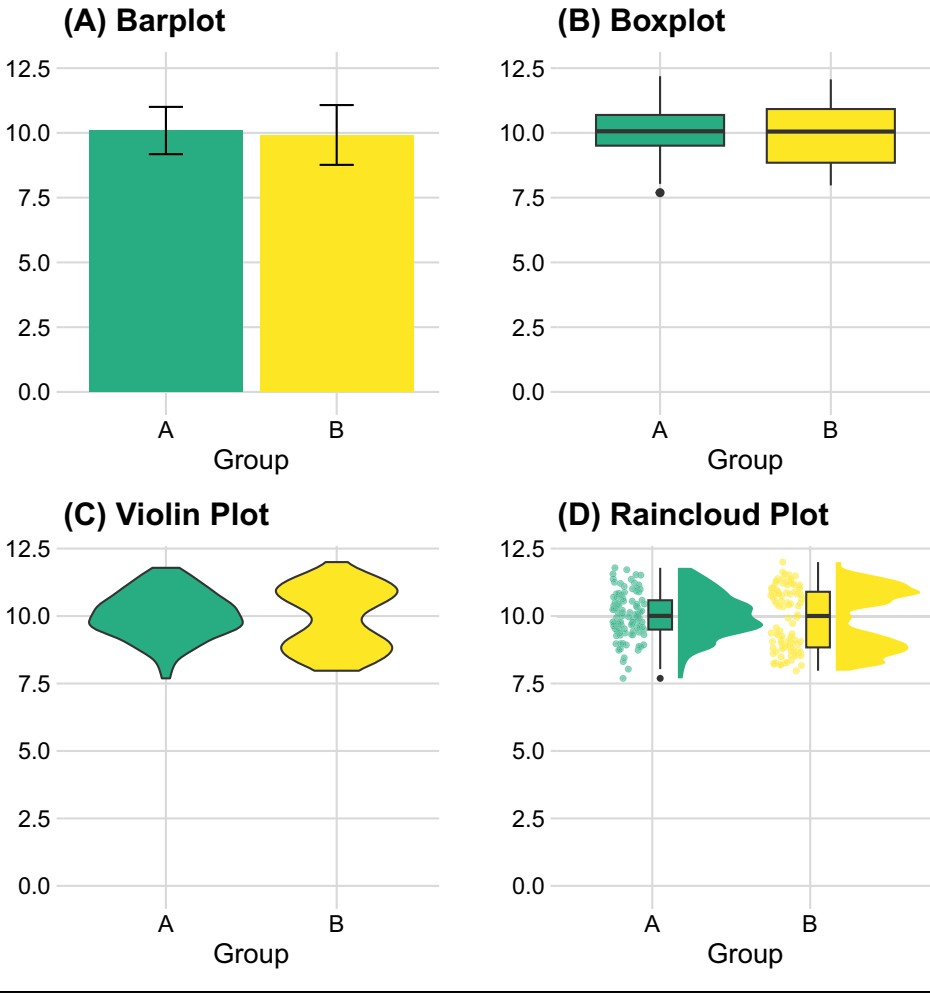

graph-gallery.com/) offers a wide range of visualization examples along with the corresponding code, making it a valuable resource for developing customized graphics. Insights from cognitive psychology and vision science can further enhance the effectiveness of data visualizations by minimizing working memory load, guiding visual attention, and leveraging familiar perceptual conventions[87]. Specifically, they emphasize that while the human visual system excels at rapidly extracting global patterns from data visualizations, it struggles with detailed comparisons between individual data points. To mitigate this, they recommend using visual channels such as position and length, which are more accurately interpreted than area or color intensity. Additionally, they advocate for the use of design elements like color highlighting, direct annotations, and proximity cues to guide viewers' attention toward the most relevant comparisons, thereby enhancing the clarity and effectiveness of data communication.

While descriptive statistics summarize key characteristics of the dataset, inferential statistics allow researchers to draw conclusions beyond the observed sample. The choice of inferential method, whether frequentist or Bayesian inference, and of the sampling design (sequential or fixed) affects how findings are interpreted and reported. The following section outlines best practices for each approach.

### Reporting inferential statistics
Inferential statistics serve as the backbone of quantitative research, enabling researchers to draw conclusions about populations based on sample data. This section outlines best practices for reporting and interpreting inferential statistics, emphasizing the importance of transparency and clarity. By adhering to these guidelines, researchers can ensure the robustness, reproducibility, and accurate interpretation of their findings.

Inferential statistics provide the foundation for drawing conclusions from psychological data, but different statistical frameworks require distinct reporting practices. Frequentist statistics remains the most widely used approach. However, it relies on concepts such as $p$-values, significance thresholds, and confidence intervals, which are often misinterpreted. Clear interpretation and adherence to reporting guidelines are therefore essential. In contrast, Bayesian inference evaluates the relative plausibility of hypotheses given the data and requires researchers to report prior distributions, Bayes factors, and credible intervals to ensure transparency and reproducibility. Sequential designs introduce additional complexities, as decisions to stop data collection affect inferential outcomes. Here, researchers must document decision boundaries, stopping rules, and adjusted estimates to account for potential biases. By understanding these differences, researchers can align their reporting practices with the underlying logic of their chosen statistical framework, ensuring clarity and interpretability for readers and future meta-analyses.

**Reporting of statistical results.** All inferential test results should be fully reported, including the key information needed to reconstruct the tests and verify the results (e.g., test statistic, degrees of freedom, prior distributions, effect size, and confidence/credible interval around the effect size). Rather than stating "*We observed a significant effect on recall accuracy* ($p < 0.05$)," which omits key statistical details, it is more informative to report: "*A significant Condition × Time interaction emerged for recall accuracy*, F(1, 54)=4.37, $p = 0.04$, *partial* $\omega^2$= 0.06 (95% CI [0.00, 0.23])." This format provides a comprehensive account of both statistical significance and effect size estimation.

**Reporting results of frequentist methods.** For frequentist methods, exact $p$-values should be reported, regardless of their statistical significance and even if they are not interpreted as a measure of evidence. Exact $p$-values can be informative for readers and can aid future meta-scientific work, such as calculating effect sizes for meta-analysis when other statistics are not available[88].

**Reporting results of Bayesian methods.** Frequentist statistics remains the dominant framework in psychology and is primarily concerned with guiding statistical decision-making through long-term error control and inductive inference from experimental data, without incorporating subjective prior information. This focus often leads to subtle and frequently misunderstood interpretations of core concepts such as $p$-values and confidence intervals[89]. For example, $p$-values are commonly misinterpreted as the probability that a hypothesis is true. Bayesian methods offer a complementary approach by explicitly incorporating prior information and enabling direct quantification of evidence for competing hypotheses.

When reporting Bayesian analyses, researchers should specify and justify prior distributions for all parameters. Priors should be described in sufficient detail to enable replication, including their mathematical form, parameter values, and rationale. Visualization of prior distributions (e.g., using the `bayestestR`[90] or `ggdist`[91] packages in R) is encouraged. Moreover, researchers should conduct and report a prior distribution sensitivity analysis, where feasible. This involves evaluating how the conclusions change across a reasonable range of priors. Tools such as `brms`[92], `rstanarm`[93], or `BayesFactor`[94] in R facilitate this process.

The models or hypotheses being compared need to be fully defined, including both the null and alternative hypotheses, when using Bayes factors. The computed Bayes factors and the method used to calculate them (e.g., default JZS prior, Savage-Dickey ratio, or bridge sampling) should be reported, including the software and version (e.g., JASP[95], `BayesFactor`[94], `bridgesampling`[96]). For example, instead of stating, "*We found a Bayes factor of 3*", a more informative report would be: "*Using a default JZS prior, we found moderate evidence for the alternative hypothesis ($BF_{10} = 3.21$), indicating that the data were approximately three times more likely under $H_1$ than under $H_0$. All priors and model specifications are provided in the Supplementary Materials.*" It is also important to correctly interpret BFs as reflecting the relative likelihood of observing the data under one hypothesis/model (e.g., $H_1$) compared to the other (e.g., $H_0$), rather than as posterior odds or absolute evidence for one of the hypotheses/models[97]. In addition, posterior distributions for parameters of interest should be included. Visualizations (e.g., density plots, interval plots) can be created using `bayestestR`[90], `ggdist`[91], or `tidybayes`[91]. Moreover, visualizations of posterior distributions should ideally be accompanied with summaries such as means, medians, and credible intervals.

Beyond Bayes factor hypothesis testing, many Bayesian analyses in psychological research focus on posterior estimation, which involves summarizing parameters with posterior distributions, credible intervals, and posterior probabilities rather than making dichotomous decisions. This estimation-focused perspective can provide a richer understanding of effect sizes and their uncertainty, complementing or even replacing formal hypothesis testing[98].

Lastly, it is important to include all settings used in the Bayesian estimation process (e.g., number of chains, number of iterations, warm-up period, thinning) to ensure reproducibility and proper model assessment. Researchers should also report convergence diagnostics to evaluate the reliability of posterior estimates. A key diagnostic is the potential scale reduction factor ($\hat{R}$), which assesses the consistency of estimates across chains and should be close to 1.00 (typically <1.01) to indicate convergence[99]. Effective sample size (ESS) should also be reported, as it quantifies the number of independent samples obtained for each parameter and informs the precision of posterior estimates.

These diagnostics are automatically provided by Bayesian modeling packages in R such as `brms`[92], `rstanarm`[100], and `cmdstanr`[101], which compute $\hat{R}$ and ESS using the `posterior` package[103]. Additionally, the `bayesplot`[102] and `ggmcmc`[104] packages allow users to generate diagnostic plots, such as trace plots, autocorrelation plots, and rank plots, which help detect problems like poor mixing or divergent transitions. For models fit using Stan-based tools (`brms`, `rstanarm`, `cmdstanr`), divergences and energy diagnostics should also be monitored and reported, as they indicate potential issues with the geometry of the posterior distribution. When convergence problems are identified, researchers should document any remedial steps taken (e.g., increasing iterations, reparameterizing the model, adjusting priors). Including such diagnostics enhances transparency and supports confidence in the robustness of Bayesian inferences.

To illustrate best practices in reporting Bayesian results[105], consider a Bayesian linear regression model estimating the effect of cognitive load on recall accuracy. One could write: "*The model was fit using the* `brms` *package (v2.21.0;[92]) with weakly informative normal priors (e.g., $\mathcal{N}(0, 1)$) on standardized coefficients. Four MCMC chains were run for 4000 iterations each (2000 warm-up), yielding 8000 post-warmup samples. Convergence diagnostics indicated that all parameters had $\hat{R} < 1.01$ and effective sample sizes (ESS) >1500, suggesting good mixing and reliable posterior estimation. The posterior distribution for the effect of high vs. low cognitive load on recall accuracy had a mean of $\beta = -0.42$, with a 95% highest density interval (HDI) of $[-0.67, -0.18]$, indicating a robust negative effect. A Bayes factor comparing the full model to a null model (excluding the load effect) was computed using bridge sampling[96] and yielded $BF_{10} = 18.7$, indicating strong evidence in favor of the model including the cognitive load effect.*"

**Reporting results of sequential designs.** When reporting results from sequential methods, researchers should include the type of test used, decision boundaries, planned stopping points, final sample size, and any adjustments made during data collection. For instance, vague reporting such as "*Data collection stopped once a significant result was observed*" should be replaced with: "*We used a sequential Bayes factor design with stopping thresholds of $BF_{10} > 6$ and $BF_{01} > 6$, computed every 10 participants. Data collection stopped at $N = 80$ when $BF_{10} = 7.2$. Sensitivity and robustness checks are reported in the Supplementary Materials.*" Researchers should also report the planned effect size, an estimated effect size with a confidence or credible interval, descriptive group means, standard deviations, and the software used. Effect size estimates in sequential designs are often biased and should be interpreted with caution. If bias adjustments were applied, they should also be reported. For more details, see the section "Conditional bias in sequential designs" below.

For the GSD design, reporting includes the a-priori planned number of looks, the actual taken (updated) looks, the correction or spending functions that are used to control for the Type I and or Type II error, the alpha and beta level at each look, the sample size at each look including the maximal sample size, and the effect size estimates and the CI of the effect (an example can be found in the work by Lakens et al. (2021)[55]). In addition, if futility bounds are used, it must be reported if they were binding or non-binding,

If a SPRT was used, the specific variant must be reported (e.g., sequential $t$-test, truncated SPRT, etc.). The alpha and the beta level, as well as the effect size or other parameters that are used to specify the alternative hypothesis should be reported. Other important aspects that need to be reported are the starting point of the SPRT (the sample size when the first look took place), the final sample size (when data collection was stopped), the final likelihood ratio, a plot to show the full likelihood progression containing all performed looks, and the effect size estimate and the CI of the effect (see Schnuerch and Erdfelder (2020)[106] for an example).

If a Sequential Bayes Factor is used, the following information should be reported: the minimum sample size, the final sample size, the upper and lower critical BFs for stopping, the type of the BF and the specification of the prior, the sequential steps at which the BF was computed, the final BF that was computed, a plot to show the BF progression, the effect size estimates, and the credible interval of the effect. For the sensitivity analysis, the alternative priors and their final BFs should be reported as well. An example can be found in the work by Schönbrodt et al.[107].

Reporting and interpreting null results. Null results play a critical role in psychological research, offering valuable insights that can shape theories and guide future studies. Moreover, reporting null results is crucial to preventing publication bias in meta analyses. However, null findings are often misinterpreted. A non-significant result in frequentist hypothesis testing should not be viewed as evidence supporting the null hypothesis. Within the Neyman-Pearson framework (see Box 1), where both null and alternative hypotheses are explicitly defined and the study is sufficiently powered, failing to reject the null leads to a decision to act *as if* the null were true. Crucially, this decision is made with the recognition that it may be incorrect in any single instance, though error rates are controlled over repeated sampling. In most cases, null results should be interpreted with caution, considering factors such as sample size, power, and the context of the research question. Proper interpretation requires understanding that a lack of statistical significance does not equate to the absence of an effect. It may instead reflect data insensitivity. For example, rather than writing, "*The effect was not significant, so we conclude there is no effect,*" it is more accurate to state: "*The effect was statistically non-significant* ($p = 0.12$), *but the 95% CI* [−0.10, 0.35] *includes values that could still be of practical importance. Thus, the data were inconclusive, and further research with increased statistical power is warranted.*"

**Equivalence testing** can complement classical hypothesis testing by allowing researchers to assess whether an observed effect is too small to be considered practically worthwhile or important[108]. Unlike traditional hypothesis testing, which mainly focuses on whether an effect exists, equivalence testing evaluates whether the effect differs meaningfully from zero within a predefined range, known as the equivalence margin. This approach is particularly valuable when the goal is to demonstrate that an effect is too small to be of practical importance, rather than simply failing to detect a significant difference. To conduct an equivalence test, researchers must establish equivalence margins based on both theoretical and practical considerations relevant to the field[109]. These margins define the range within which the effect size is deemed negligible. The test then determines whether the confidence interval around the observed effect falls entirely within this margin, indicating that the effect is equivalent to zero in a practical or theoretical sense (see the left part of Box 4). To maintain a type-I error rate of at most $\alpha$, equivalence testing should use a $(1−2\alpha) \times 100\%$ confidence interval (e.g., 90% for $\alpha = 0.05$). Alternatively, equivalence can be assessed via the two one-sided tests (TOST) procedure, which provides an equivalent formal test of the null hypothesis that the true effect lies outside the pre-specified equivalence bounds[109].

Although pre-specifying equivalence margins is preferred to avoid bias, researchers may also perform post hoc equivalence tests after non-significant results to aid interpretation. In such cases, the chosen margin should be explicitly justified, and the exploratory nature of the analysis should be acknowledged, as post hoc margin selection reduces inferential stringency[110] and conditional equivalence tests can introduce biases[111] (for more details, see below). To enhance transparency, researchers are encouraged to report results across a range of plausible margins, demonstrating that conclusions are not unduly dependent on arbitrary choices. Moreover, equivalence margins should be reported for both significant *and* non-significant findings. Finally, reports should clearly state the selected margins and whether the confidence interval for the observed effect lies entirely within them, confirming equivalence.

Equivalence testing is not merely a follow-up to non-significant results, as proposed in the framework of *conditional equivalence testing*[112], but also functions as a primary inferential approach in studies aiming to demonstrate that an effect is sufficiently small to be considered practically or theoretically negligible (e.g., in non-inferiority research). Importantly, reporting equivalence tests only after observing non-significant results can introduce bias by applying different inferential standards to significant versus null findings[111]. Since equivalence testing requires larger sample sizes to achieve adequate power than traditional significance testing[113], post hoc applications should be interpreted with caution, particularly when such tests were not preregistered or planned a priori.

An insufficient example of reporting results from equivalence testing would be: "*We observed no significant effect of load on RT* ($p = 0.92$). *This indicates that memory load had no effect on participants' RT.*" Instead, researchers should provide further details about the equivalence test to contextualize null findings: "*We observed no significant effect of load on RT, $F(1,87) = 0.01$, $p = 0.92$, partial $\omega^2 = 0.00$ (95% CI [0.00, 0.01]). We further ran an equivalence test using the Two One-Sided Tests (TOST) procedure with ±0.20 as our smallest effect size of interest. TOST evaluates whether the observed effect falls entirely within a predefined equivalence margin by testing two complementary null hypotheses — one that the effect is smaller than the lower bound, and one that it is larger than the upper bound. If both null hypotheses are rejected, the effect is deemed statistically equivalent to zero within the specified bounds. We defined the smallest effect size of interest as Cohen's $d = ±0.20$, consistent with established conventions for small effects in cognitive psychology. The observed standardized mean difference was $d = 0.02$, with a 90% confidence interval of [−0.16, 0.20]. Because this interval fell entirely within the equivalence bounds of −0.20 and +0.20, the effect was eligible for equivalence testing. The first one-sided t-test evaluated whether the observed effect was significantly greater than the lower bound of the smallest effect size of interest (−0.20). This test yielded $t(87) = 2.08$, $p = 0.020$. The second one-sided t-test evaluated whether the observed effect was significantly less than the upper bound of the smallest effect size of interest (+0.20), yielding $t(87) = −1.67$, $p = 0.049$. Because both null hypotheses were rejected at the $\alpha = 0.05$ level, we concluded that the effect of memory load on RT was statistically equivalent to zero within the specified equivalence bounds.*" This not only avoids misinterpretation but adds inferential value to null results. In general, null findings should be contextualized rather than dismissed.

**Bayesian methods** also offer a valuable approach for interpreting null results by quantifying the relative strength of evidence for the null versus the alternative hypothesis, providing a nuanced understanding of the data (see the right panel of Box 4). A BF close to 1 indicates that the data do not strongly favor either hypothesis, suggesting insufficient evidence. Typically, a BF greater than 3 or less than 1/3 is interpreted as moderate evidence for $H_0$ or $H_1$, respectively, while BFs exceeding 10 or below 1/10 are considered strong evidence[114,115]. Although these suggested thresholds can provide a useful starting point for interpreting Bayes Factors, descriptions of the magnitude of relative evidence for a given hypothesis should be guided by the specific research context[116]. Altogether, this approach allows researchers to assess the relative support for the null hypothesis in comparison to the alternative hypothesis, rather than simply failing to reject it. When reporting BFs, it is crucial to detail both the null and alternative hypotheses, specifying the statistical hypotheses being compared and the prior distribution used. It is also important to interpret BFs correctly — as a measure of relative evidence for one hypothesis compared to the other — rather than as absolute evidence for one of the two hypotheses. For a more comprehensive discussion on using Bayes Factors, see Tendeiro et al. (2024)[97].

As with conventional analyses, researcher flexibility can influence outcomes in both equivalence testing and Bayesian hypothesis testing. In equivalence testing, pre-specifying equivalence bounds before data analysis improves inferential stringency and prevents their selective specification after the results are known[110]. If pre-specification is not feasible, researchers should report results across a range of bounds to demonstrate that conclusions are robust and not unduly influenced by arbitrary decisions. Failing to do so may lead to "reverse p-hacking," where flexible analysis practices are used to assert the absence of an effect, undermining the integrity of the findings[109]. As Bayes factors can be sensitive to the prior (see Box 2), Bayesian hypothesis testing is similarly susceptible to "B-hacking," where researchers might manipulate prior distributions to achieve desirable results[117]. To prevent this, priors should be pre-specified, even if default priors are used. This transparency ensures that the results are not a product of selective reporting but reflect genuine evidence for or against the hypotheses under investigation. Pre-specification and transparency are crucial for maintaining the credibility of equivalence and Bayesian testing. If

## Box 4 | Resources for the interpretation of null results

**Interpreting equivalence tests**

The equivalence test tests if the null hypothesis that an effect is at least as small as a pre-specified lower equivalence margin *l* or at least as large as a pre-specified upper equivalence margin *u* can be rejected. These margins define the range within which the effect size is deemed negligible. It uses interval hypothesis testing to test the hypothesis that the effect falls in a range around 0.

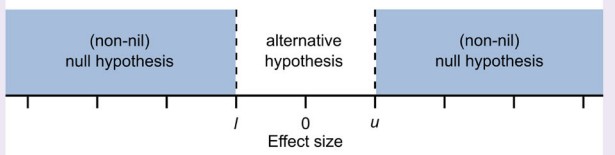

**Further reading:**
Lakens, D., Scheel, A. M., & Isager, P. M. (2018). Equivalence Testing for Psychological Research: A Tutorial. *Advances in Methods and Practices in Psychological Science*, *1*(2), 259–269. https://doi.org/10.1177/2515245918770963

**Interpreting Bayes factors**

As the Bayes factor (BF) moves away from 1, which represents equal support for the $H_0$ and $H_1$, the evidence increasingly favors one hypothesis over the other. Bayes factors in the range of 1 to 3 are considered weak (barely worth noting), those between 3 and 10 indicate moderate evidence, and values exceeding 10 reflect strong evidence.

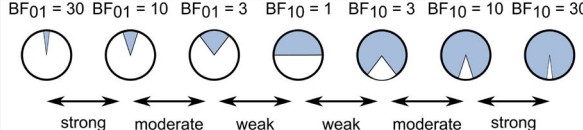

**Further reading:**
van Doorn, J., van den Bergh, D., Böhm, U., Dablander, F., Derks, K., Draws, T., Etz, A., Evans, N. J., Gronau, Q. F., Haaf, J. M., Hinne, M., Kucharský, Š., Ly, A., Marsman, M., Matzke, D., Gupta, A. R. K. N., Sarafoglou, A., Stefan, A., Voelkel, J. G., & Wagenmakers, E.-J. (2021). The JASP guidelines for conducting and reporting a Bayesian analysis. *Psychonomic Bulletin & Review*, *28*(3), 813–826. https://doi.org/10.3758/s13423-020-01798-5

priors have not been specified, then sensitivity analyses should be run using a range of prior distributions.

Lastly, **sequential tests** also allow to draw conclusions about the null and the alternative hypothesis. Specifically, the SPRT and SBF are suitable choices as they allow for the direct accumulation of evidence supporting the null hypothesis. Multiple aspects should be considered for a convincing null result: the expected effect size of interest, which defines the alternative hypothesis, should be the smallest effect of interest or the effect of a safe guard power analysis[41], the decision boundaries need to be set adequately so that the power is sufficient, and in order to accept the null hypothesis the data collection can only be stopped because a decision boundary is reached and not because the resources are depleted beforehand. On average, gathering sufficient evidence to accept the null hypothesis requires more data points than for the alternative hypothesis[118]. Hence, examining the ASN prior to testing is advisable. If any of the above requirements are not fully met, the results may still be reported; however, the strength of evidence and interpretability are substantially reduced. Similarly, GSDs can allow researchers to accept the null hypothesis in a manner akin to treating it as true-particularly when the final stage of the design is reached and statistical power is sufficiently high[119]. However, it is crucial to distinguish this from stopping early for futility. Terminating a study at an interim stage to conserve resources does not constitute accumulating adequate evidence in favor of the null hypothesis. In such cases, equivalence testing based on the smallest effect size of interest, as previously discussed, is appropriate[109,119]. Regardless of the test framework used, all relevant aspects of the sequential design need to be transparently reported, ideally accompanied by pre-registration, to ensure the evidence presented is compelling and credible. For example, "*We used a sequential design and stopped at N = 60*" is insufficient. Instead, one might report: "*We preregistered a group sequential design with three interim looks, using an O'Brien-Fleming alpha-spending function to maintain the familywise Type I error rate at .05. The study was stopped at the second interim analysis (N = 60) after reaching the critical boundary.*"

Reporting results of meta-analyses. The data used to perform a meta-analysis should be made available using an accessible file format (e.g., .csv file). Like data from primary studies, open data can be used to verify and better understand analyses for future work. However, there is an important distinction between meta-analysis data and primary human research data. Maintaining research participant anonymity can be a justifiable reason, in

many cases, for not sharing raw data underlying an article's reported analysis. But as the data used to perform meta-analysis is already publicly available summary statistics or based on summary statistics derived from unpublished data, there no risk that individuals can be identified. While summary effect sizes are often the focus of meta-analysis outcomes, heterogeneity can also be highly informative. As commonly reported heterogeneity measures (e.g., $I^2$, $Q$, $\tau^2$) have both advantages and limitations, it has been recommended to report $I^2$ along with other measures[120]. In addition, authors should use established meta-analysis reporting guidelines, such as PRISMA[121,122] or MOOSE[123].

A common misconception in meta-analysis is that measures of funnel plot asymmetry, such as Egger's regression test or visual inspection of the funnel plot, represent publication bias. While funnel plot asymmetry measures can certainly *encompass* publication bias, other elements associated with small studies can also be included[124]. Thus, it is more accurate to report funnel plot asymmetry as "small study bias". There are several options available for detecting and/or correcting for publication bias (e.g., selection models). However, such methods operate under certain sets of assumptions and are associated with various limitations, which should be carefully considered when planning analyses and interpreting results[44].

**Effect sizes: estimation, interpretation, and context.** Statistical results should always be accompanied by estimates of effect sizes and a measure reflecting the uncertainty surrounding their point estimates, such as confidence or credible intervals[125]. A *frequentist confidence interval* is a range of values that, under repeated random sampling, would contain the true population parameter in a specified proportion of cases (e.g., 95% of the time for a 95% confidence interval). The 95% refers to the long-run frequency of intervals containing the true parameter if the procedure were repeated many times, not the probability that the true parameter lies within the calculated interval. *Bayesian credible intervals*, in contrast, represent the range of values within which the true effect size lies with a certain degree of probability, given the prior distribution and observed data. While statistical tests indicate whether an effect exists, effect sizes quantify its magnitude. A key advantage of most effect sizes is their independence from a study's sample size. However, the uncertainty in these estimates is still influenced by sample size, with smaller samples leading to greater uncertainty. Therefore, it is essential to report

appropriate measures of uncertainty and interpret observed effect sizes with this in mind.

Effect sizes can be unstandardized, such as the difference between two means, or standardized, which makes them independent of a study's measures. This standardization is crucial for comparing findings across different studies. Effect sizes are particularly important for meta-analyses, where they are used to aggregate evidence from multiple studies, leading to more precise estimates of population parameters.

Choosing an appropriate effect size for a given statistical test can be challenging. This difficulty stems either from the multitude of available effect size metrics for relatively simple analyses (e.g., ANOVA) or from the structural complexity of certain models (e.g., multilevel or generalized models). To help researchers navigate these decisions, Table 5 summarizes commonly used effect size measures across popular statistical model families. In addition to defining each metric, the table identifies freely available software tools and packages that can be used to estimate these values and, where possible, compute associated confidence intervals. This practical information is intended to support the transparent and consistent reporting of effect sizes in quantitative research.

Conditional bias in sequential designs. Although effect size estimates should always be reported in sequential designs, they must be interpreted with caution as they are, on average, less precise and tend to be biased[126]. Sequential testing is designed to reach a decision between two hypotheses as efficiently as possible, often resulting in smaller sample sizes on average compared to fixed sample designs[127]. However, large samples are required for accurate effect size estimates, making sequential testing less suitable for precise estimation due to its inherently lower precision. The high efficiency of sequential testing is rooted in the fact that the evidence in the data is used for the decision when to stop the data collection, which introduces a systematic bias: Smaller sample sizes tend to overestimate the true effect, and larger sample sizes tend to underestimate the true effect. While the unconditional bias can be substantially reduced in meta-analyses by weighting estimates according to the respective sample size[107,118], the conditional bias in single-study estimates is more difficult to address[126]. Moreover, the conditional bias is particularly impactful in smaller samples, where the deviation from the true effect is most pronounced. Bayesian statistics, with its concept of prior-induced shrinkage, provides an intuitive approach for addressing the positive conditional bias in early stoppings, because the influence of the prior is stronger in smaller samples[128]. In this approach, the prior distribution represents a plausible range of effect sizes within the specific research context, adjusting the estimated effect size, especially in small samples and when the estimate is unexpectedly large and therefore also less likely to be accurate. More complex approaches also exist to reduce the conditional bias for other sequential designs[126]. In cases where the software does not offer correction for conditional bias, reports should explicitly acknowledge this limitation. For example: "*The estimated effect size was d = 0.56, 95% CI [0.19, 0.93]. However, this value may be upwardly biased due to early stopping in the sequential design. Because our software did not implement conditional bias correction, we encourage caution in interpreting the magnitude of this estimate.*" Systematic conditional biases may arise in individual sequential samples, especially when effect size estimates are unexpectedly large. Transparent reporting of these potential biases is critical for accurate interpretation. Calculating confidence intervals poses a similar challenge. For GS designs, repeated confidence intervals[129,130] can be calculated with the `rpact` package[131]. These confidence intervals are corrected for the multiple-look problem and are constructed by inverting the sequential test, meaning that each interval contains the parameter values that are not rejected by the sequential test at a specific stage. Although the literature contains promising approaches to also calculate confidence intervals for SPRTs[132,133], to date, such methods are not implemented in available software packages. It is possible to calculate a naive confidence interval (using methods designed for fixed designs) for the SPRT, as implemented in the `sprtt` package[134]. However, such naive CIs result in

coverage probabilities lower than specified; for instance, a 95% confidence interval covers the true effect only 93% of the time in simulations[118].

Interpreting effect sizes: beyond Cohen's benchmarks. Cohen's framework is a widely used guide for categorizing effect sizes, providing standardized benchmarks for small ($d = 0.20$), medium ($d = 0.50$), and large ($d = 0.80$) effects across various fields of research. What is less known is that Cohen explicitly acknowledged that these thresholds were arbitrary and meant to provide rough guidelines rather than definitive cut-offs[135]. Recent research has highlighted the need for its revision based on empirical findings (in psychology)[136–142]. For instance, a correlation of $r = 0.30$, which is a medium effect according to Cohen's thresholds, has been argued to represent a large effect in some research contexts, particularly in psychology, where effect sizes tend to be smaller on average[136]. One might report: "*Participants' self-esteem was positively associated with extraversion (r = 0.20, 95% CI [0.05, 0.34], p = 0.01), an effect size that is considered moderate according to current guidelines in personality and individual differences research[137].*" Moreover, the practical or real-world significance of an effect depends on its context and frequency. For example, a small but frequent effect might have a greater cumulative impact than a larger, rarer effect[136]. In such cases, the context-specific consideration of unstandardized effects may provide additional useful information[143]. For instance: "*Participants receiving the health intervention increased their usual walking speed by 0.12 m/s relative to controls (d = 0.35), surpassing the 0.10 m/s benchmark for a clinically substantial change in older adults[144].*" Critiques suggest that effect size benchmarks may require domain-specific adjustments to more accurately reflect the characteristics of the phenomena, and highlight the need for a deeper understanding of variations in effect sizes driven by study-specific factors, such as differences in materials or dose-dependent relationships, which can be modeled in meta-analyses[136,145]. In summary, when empirically established benchmarks are available within a research field, they should be reported to facilitate more nuanced discussions within the field. While Cohen's thresholds can be a valuable framework for communicating effect sizes across disciplines, they should be applied and interpreted with careful consideration of the specific context and limitations. An alternative approach is to create a field-specific effect size distribution that accounts for publication bias[142].

### Common reporting errors and how to avoid them

This section aims to elucidate common errors in statistical reporting and provide guidance on how researchers can improve their practices to maintain the integrity of their findings. For an extensive review, see Lydersen (2025)[146]. Overviews of questionable research practices beyond reporting are provided for frequentist fixed-sample designs[147], frequentist sequential designs (SPRT)[148], and the misuse of Bayes factors[97].

**Biased effect size estimates.** Cohen's $d$ and partial $\eta^2$ are two of the most common effect size estimates reported in psychological research[149]. However, both effect size estimates are positively biased, meaning that they tend to overestimate the population effect size[150,151], especially when sample sizes are small. Therefore, it is recommended to use bias-corrected effect size estimates: Hedges' $g$ for t-tests[152] and $\epsilon^2$ [153] or $\omega^2$ [154] for ANOVAs (see Table 6).

**Differences of differences.** Differences of differences need to be formally tested. For example, a non-significant correlation and significant correlation are not necessarily "different". The "Cocor" package[155] can test this, for example. Likewise, moderation effects need to be formally tested with interaction effects and not assumed based on findings from separate statistical models.

**Hypothesizing After Results are Known (HARKing).** HARKing is a questionable research practice in which post hoc hypotheses are presented as if they were formulated a priori[156]. This may be done

**Table 5 | Overview of effect sizes for common statistical models**

| Statistical Model | Effect Size | Description | R Packages |
|---|---|---|---|
| **General Linear Models** | | | |
| Regression | $\beta$ | Standardized regression weight[135] | `lm.beta` |
| | partial $R^2$ | Unique variance explained by predictor[135] | `rsq`, `effectsize` |
| | Cohen's $f^2$ | Local ES for a predictor[135] | `effectsize` |
| *t*-test | Cohen's $d$ | Uncorrected standardized mean difference[135] | `effectsize`, `TOSTER` |
| | Hedges' $g$ | Corrected standardized mean difference[193] | `effectsize` |
| | Glass's Δ | Using one group's SD[194] | `effsize` |
| ANOVA/ANCOVA | Cohen's $f$ | Dispersion of means across groups[135] | `effectsize` |
| | $\eta^2$ | Eta squared (uncorrected correlation ratio)[195] | `effectsize` |
| | $\omega^2$ | Less biased alternative to $\eta^2$[196] | `effectsize` |
| MANOVA | $\eta^2_{partial}$ | Multivariate eta squared[197] | `heplots` |
| **Generalized Linear Models** | | | |
| Logistic/Ordinal Regression | Odds Ratio | Comparison of outcome odds[198] | `epiR`, `epitools` |
| | Std. logit $\beta$ | Log-odds change per 1 SD predictor? | `effectsize`, `logistf` |
| | McFadden's $R^2$ | Likelihood-based pseudo-$R^2$[199] | `pscl` |
| Poisson/Neg. Binomial | Rate Ratio | Relative event rate[200] | `MASS`, `epitools` |
| | Std. log-rate $\beta$ | Log-rate change per 1 SD predictor[200] | `effectsize`, `MASS` |
| | McFadden's $R^2$ | Pseudo-$R^2$ for count models? | `pscl` |
| Multinomial Logistic | Relative Risk Ratio | Probability ratio vs. baseline[201] | `nnet`, `brglm2` |
| | McFadden's $R^2$ | Likelihood-based pseudo-$R^2$[199] | `pscl::pR2` |
| Beta Regression | Odds Ratio | Comparison of outcome odds[202] | `betareg` |
| | Pseudo-$R^2$ | Likelihood-based pseudo-$R^2$[202] | `betareg` |
| **Mixed-Effect Models** | | | |
| Linear Mixed-Effects | $\beta$ | Fixed-effect coefficient[203] | `lme4`, `lmerTest` |
| | Cohen's $f^2$ | Local ES for fixed predictor[135] | `effectsize` |
| | $R^2_{marginal}$ | Variance explained by fixed effects[204] | `performance`, `MuMIn` |
| | $R^2_{conditional}$ | Variance explained by fixed + random effects[204] | `performance`, `MuMIn` |
| GLMM | Odds Ratio | Comparison of outcome odds[198] | `lme4`, `glmmTMB` |
| | Rate Ratio | Relative event rate[200] | `glmmTMB` |
| | $R^2_{marginal}$ | Variance explained by fixed effects[204] | `performance`, `MuMIn` |
| | $R^2_{conditional}$ | Variance explained by fixed + random effects[204] | `performance`, `MuMIn` |
| Random Effects | ICC | Proportion of variance due to grouping[204] | `psych`, `irr` |
| | VPC | Variance partition coefficient[203] | `lme4`, `performance` |
| **Structural Equation Models** | | | |
| Path Analysis | $\beta$ | Standardized path coefficient[205] | `lavaan` |
| CFA | $\lambda$ | Standardized factor loading[206] | `lavaan` |
| Latent Growth Modeling | Effective curve reliability | Slope variance scaled by error[207] | `lavaan` |
| | $\beta$ | Path coefficient for latent change[208] | `blavaan` |
| **Survival Analysis** | | | |
| Cox PH Model | HR | Hazard ratio (event likelihood over time)[209] | `survival` |
| Kaplan-Meier | Median survival diff. | Median survival comparison[209] | `survminer` |
| AFT Models | Time Ratio (TR) | Relative time to event[210] | `survival` |
| **Nonparametric Models** | | | |
| KDE | Bandwidth ES | Effect of smoothing bandwidth[211] | `stats::density` |
| Mann-Whitney U | Rank-Biserial Corr. | Rank-based association[212] | `rcompanion`, `coin` |
| Kruskal-Wallis | $\eta^2_{ranks}$ | Adaptation of eta-squared for ranks[213] | `rstatix` |
| **Multidimensional Scaling (MDS)** | | | |
| Classical MDS | Stress Value | Goodness-of-fit of distance data[214] | `cmdscale`, `smacof` |
| Nonmetric MDS | Stress Value | Rank-order fit in MDS[214] | `smacof` |

**Table 5 (continued) | Overview of biased effect size estimates, recommended alternatives, and supporting software**

| Statistical Model | Effect Size | Description | R Packages |
|---|---|---|---|
| **Cluster Analysis** | | | |
| K-means | Silhouette Coeff. | Cluster separation/fit[215] | `cluster`, `factoextra` |
| Hierarchical | Cophenetic Corr. | Correlation between distances and tree[216] | `stats`, `dendextend` |
| Gaussian Mixture | BIC | Model quality vs. parameters[217] | `mclust` |
| **Network Analysis** | | | |
| Social Networks | Density/Centrality | Node connections/influence[218] | `igraph` |
| Psychometric Networks | Node/Bridge Strength | Connectivity and clustering[219] | `qgraph`, `bootnet` |

Primary effect sizes quantify interpretable parameter magnitudes; model fit indices (e.g., $R^2$, BIC) are complementary.

**Table 6 | Overview of biased effect size estimates, recommended alternatives, and supporting software**

| Effect size estimate | Description | Bias | Recommendation | R Package |
|---|---|---|---|---|
| Cohen's $d$ | Commonly used for measuring standardized mean difference | Positive bias in small samples | Hedges' $g$[152] | `effectsize`[220] |
| $\eta^2$ | Measure of variance explained in ANOVA | Positive bias in small samples | $\epsilon^2$ [153], $\omega^2$ [154] | `effectsize`[220] |

deliberately to craft a more compelling or simplified narrative or unintentionally through cognitive biases like hindsight and confirmation bias. Regardless of intent, HARKing misrepresents the research process and can inflate the familywise Type I error rate within a frequentist statistical framework[147,156]. Exploratory analyses typically involve examining many patterns without prespecified hypotheses. When statistical significance alone determines which results are reported, and no appropriate correction is made for the large number of tests conducted, the risk of false positives increases substantially. This risk is often underestimated because random variability and the ease with which statistical significance can occur in exploratory contexts with many potential hypotheses are not adequately accounted for. Moreover, post hoc rationalizations can be mistakenly perceived as preplanned. Preregistration mitigates these issues by requiring a clear specification of hypotheses in advance, fostering transparency between exploratory and confirmatory work. Importantly, the problem with HARKing lies not in exploratory research itself, which is vital to discovery, but in misrepresenting it as confirmatory.

**Misinterpretation of frequentist confidence intervals.** One of the most common misinterpretations of confidence intervals is the belief that a 95% CI means there is a 95% probability that the true parameter lies within the observed interval. This interpretation incorrectly treats the interval as a probabilistic statement about the specific sample. However, in the frequentist framework, CIs reflect properties of the statistical procedure, not the individual sample. A 95% CI means that if we were to repeat the same experiment an infinite number of times, using the same procedure each time, approximately 95% of the resulting intervals would contain the true parameter value. It is therefore a statement about the long-run performance of the method, not about the probability of the parameter being in any single observed interval. Understanding this distinction is essential for accurate interpretation and transparent reporting. While CIs provide useful information about estimate precision and uncertainty, they do not quantify the probability of a hypothesis or parameter being true, nor do they offer direct evidence for or against a hypothesis. Further discussion of these issues can be found in several critical reviews of statistical interpretation practices[157,158]. However, there is an interval that can be interpreted in this probabilistic way: in Bayesian statistics, such an interval can be constructed by making additional assumptions about plausible effect sizes, expressed in the form of a prior distribution. These intervals are usually referred to as credible intervals. Practical guidance on constructing and interpreting credible intervals is available in applied tutorials and methodological evaluations[159,160].

**Misinterpretation of *p*-values.** *P*-values are frequently misinterpreted, which can lead to erroneous conclusions and undermine cumulative science. A common but relatively minor misinterpretation is describing non-significant results as "trending toward significance." This kind of conclusion, which implies that the test would have been more significant if more data were collected, can be misleading as the collection of additional data may also lead to *p*-values "trending away" from significance[161]. Instead, one can write that a *p*-value was on the threshold of statistical significance.

More severe and widespread misinterpretations include treating the *p*-value as the probability that the null hypothesis is true, interpreting it as evidence for the alternative hypothesis in comparison to the null hypothesis, or as the probability that the observed data were generated purely by chance[6,157]. Such interpretations conflate frequentist and Bayesian concepts and overstate the evidential meaning of *p*-values. Correctly understood, a *p*-value represents the probability of observing data at least as extreme as those obtained, assuming that $H_0$ and its associated model are true. Clear communication of this definition is essential to avoid misinterpretation.

**Multiple testing and alpha inflation.** Not adjusting alpha levels for families of multiple tests can increase false positives. Although there is much discussion on what constitutes a 'family'[162], the correction method and approach must be clearly explained (i.e., for how many tests a correction was performed). In addition, some correction methods for multiple tests are more conservative than others. Therefore, pre-specifying the correction approach and number of corrections can increase the falsifiability of a test. For example: "*We preregistered that Holm-Bonferroni correction would be used across four primary comparisons to control the familywise error rate at $\alpha = 0.05$.*"

**Post hoc power analysis ("observed power").** A common reporting error involves the use of post hoc power analysis, also referred to as *observed power*. This technique entails calculating the statistical power of a test after observing the results, typically using the effect size estimated from the data. Although it is often presented in an attempt to justify non-significant findings or to assess the adequacy of a study's sample size, post hoc power analysis is fundamentally flawed[163]. The key issue is that observed power is mathematically tied to the *p*-value; it does not provide any information beyond what is already conveyed by the result of the hypothesis test. When a result is statistically non-significant, the observed power will inevitably be low, and conversely, it will be high for statistically significant results. In this sense, post hoc power simply rephrases the outcome of the significance test without offering any meaningful additional insight.

Rather than relying on post hoc power calculations, researchers should focus on reporting the precision of their estimates through confidence intervals around effect sizes[164]. In cases of non-significant findings, equivalence testing or Bayesian approaches can provide more informative interpretations regarding the presence or absence of meaningful effects. Another useful approach is *sensitivity analysis*, which is often mistakenly conflated with post hoc power analysis. However, the two differ conceptually and inferentially: while post hoc power analysis uses the observed effect size and thus reiterates the *p*-value[33], sensitivity analysis estimates the smallest effect size that a study could have detected with reasonable power, given its sample size and alpha level[165]. This makes sensitivity analysis particularly valuable when interpreting null results from secondary or archival data. Ultimately, the validity of statistical inferences should rest on well-documented a priori power analyses or transparent alternatives like sensitivity analysis and not retrospective justifications that conflate statistical power with observed outcomes.

**Selective application of inferential frameworks**. A subtle but increasingly problematic reporting error involves the selective application of different inferential systems within the same study, most commonly using significance testing for most analyses but then applying Bayes factors or equivalence tests specifically to non-significant results. While this practice may appear methodologically sophisticated, recent simulation work demonstrates that such selective application systematically overestimates evidence favoring the null hypothesis when true population effects exist[166]. This occurs because non-significant results constitute a non-random subset of all possible outcomes, and conditioning analytical choices on initial test results creates "inadvertent cherry-picking" of inferential approaches[111]. Rather than mixing frameworks based on the convenience of particular results, researchers should either commit to a single inferential system throughout their analysis or, if reporting multiple approaches, consistently base their conclusions on one framework regardless of which specific results it favors. For example, instead of stating "*We used t-tests for our analyses and additionally computed Bayes factors for the non-significant results*", a more appropriate approach would be "*We conducted both frequentist and Bayesian analyses for all comparisons but base our inferences exclusively on the Bayesian results*" or "*We used frequentist methods throughout but computed Bayes factors for all tests to provide additional interpretive context.*"

**Visual misrepresentation of data**. Visual descriptions of data can provide insights into key aspects of data, including data distribution, linearity of data, outliers, patterns of missing data, and the validity of assumptions required for various statistical analyses[167]. Despite their importance, many published studies continue to rely on inadequate or potentially misleading visualization techniques. A common example is the widespread use of bar plots with error bars to summarize group-level means, which conceal the underlying distribution and variability of the data[82]. Such practices may obscure key information, including multimodality, skewness, and the presence of influential data points, thereby reducing the transparency and interpretability of findings. To promote clearer and more informative reporting, it is recommended to use visualizations that more accurately reflect the structure of the data. Overall, adopting more informative and transparent data visualization practices enhances reproducibility and facilitates critical evaluation of reported findings. Several guidelines and editorials from leading journals have emphasized the importance of moving away from summary-only plots toward richer visualizations that display individual data points[84,168,169]. Examples of such recommended practices are illustrated in Fig. 1.

## Conclusions

It is crucial that scientific articles provide a comprehensive account of how a research project was conducted. Ensuring that sufficient information is provided at the point of submission can facilitate more effective and collaborative scientific dialogue. Beyond enhancing understanding, transparent reporting offers the added benefit of more easily facilitating future meta-scientific work. By ensuring that articles are fully self-contained, authors can not only bolster the article's credibility but also advance meta-scientific work that relies on comprehensive and transparent reporting.

Transparent statistical reporting is essential for ensuring the credibility, reproducibility, and interpretability of psychological research. At the planning stage, researchers should clearly define their hypotheses, specify analysis plans in advance whenever possible, and justify their sample size decisions based on statistical power considerations or Bayesian precision requirements. Preregistration of hypotheses and analysis plans can enhance transparency and reduce researcher degrees of freedom, though it should be acknowledged that preregistration alone does not guarantee better research quality. When reporting results, researchers must provide sufficient detail for reproducibility, including exact test statistics, effect sizes with confidence or credible intervals, and the rationale behind statistical choices. Frequentist analyses should include precise *p*-values and effect sizes, Bayesian analyses must report priors and Bayes factors, and sequential designs require explicit documentation of stopping rules and decision boundaries. The interpretation of null results demands particular care, as failing to reject the null hypothesis does not constitute evidence in its favor; equivalence testing and Bayesian approaches can provide more informative alternatives. Avoiding common errors, such as selective reporting, inappropriate multiple comparisons corrections, and misinterpretation of *p*-values, is crucial for maintaining scientific integrity.

Ultimately, transparent reporting not only strengthens individual studies but also facilitates cumulative knowledge building by enabling more robust meta-analyses and systematic reviews. By adhering to the principles and examples provided in the present paper, researchers can contribute to a more rigorous and reliable psychological science.

## Reporting summary

Further information on research design is available in the Nature Portfolio Reporting Summary linked to this article.

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

## Acknowledgements
We thank the members and master students of the Analysis and Modeling of Complex Data Lab at the Johannes Gutenberg University Mainz for their valuable feedback on an earlier version of this manuscript. This work was supported by the Carl Zeiss Foundation [P2019-01-003; 2021-2026] and by the Research Council of Norway [324783] via their salary financing. The funders had no role in the preparation of the manuscript or the decision to publish.

## Author contributions
A.-L.S. and D.S.Q. conceptualized the manuscript. A.-L.S., M.S., H.K., and D.S.Q. wrote parts of the initial draft and reviewed the manuscript.

## Competing interests
Anna-Lena Schubert and Daniel Quintana are Editorial Board Members for Communications Psychology. They were not involved in the editorial review of, nor the decision to publish this article. All other authors declare no competing interests.
