## [Transparent Peer Review file · Communications Psychology]

Improving Statistical Reporting in Psychology

Corresponding Author: Professor Anna-Lena Schubert

Version 0:

Decision Letter:

Dear Professor Schubert,

Thank you for your patience during the peer-review process. Your manuscript titled "Improving Statistical Reporting in Psychology" has now been seen by 3 reviewers, and I include their comments at the end of this message.

The reviewers are in principle enthusiastic about your work. However, they also mention a number of concerns. We are very interested in the possibility of publishing your manuscript in *Communications Psychology*, but would like to consider your response to these concerns in the form of a revised manuscript before we make a decision on publication.

To aid you with that task, I have included a marked-up version of your manuscript. Furthermore, to enhance the practical value of the piece, we ask you to ensure that each recommendation is accompanied by a reference to appropriate, ideally free/open source software.

In sum, we invite you to revise your manuscript taking into account all reviewer and editor comments.

EDITORIAL POLICIES AND FORMATTING

*** TRANSPARENT PEER REVIEW:** *Communications Psychology* uses a transparent peer review system. This means that we publish the editorial decision letters including Reviewers' comments to the authors and the author rebuttal letters online as a supplementary peer review file. We publish these records for all accepted manuscripts. However, on author request, confidential information and data can be removed from the published reviewer reports and rebuttal letters prior to publication. If your manuscript has been previously reviewed at another journal, those Reviewers' comments would not form part of the published peer review file.

If you have any questions about any of our policies or formatting, please don't hesitate to contact me.

Please use the following link to submit your revised manuscript and a point-by-point response to the referees' comments (which should be in a separate document to any cover letter):

Link Redacted

We hope to receive your revised paper within 12 weeks; please let us know if you aren't able to submit it within this time so that we can discuss how best to proceed. If we don't hear from you, and the revision process takes significantly longer, we may close your file.

We understand that due to the current global situation, the time required for revision may be longer than usual. We would appreciate it if you could keep us informed about an estimated timescale for resubmission, to facilitate our planning. Of course, if you are unable to estimate, we are happy to accommodate necessary extensions nevertheless.

Please do not hesitate to contact me if you have any questions or would like to discuss these revisions further. We look forward to seeing the revised manuscript and thank you for the opportunity to review your work.

Best regards,

Jennifer Bellingtier, PhD
Senior Editor
Communications Psychology

REVIEWERS' EXPERTISE:

Reviewer #1 statistics
Reviewer #2 statistics
Reviewer #3 statistics

REVIEWERS' COMMENTS:

Reviewer #1 (Remarks to the Author):

This paper offers clear and constructive guidance on statistical reporting in psychology. It will be very useful for researchers, reviewers, and editors, but perhaps most importantly for students and early-career researchers who are trying to grasp the evolving norms of a rapidly improving field. I also expect this paper to be used in teaching, in addition to informing research practices.

I enjoyed reading it and appreciate its clarity and thoughtful signposting to additional resources where appropriate. The example sentences provided are especially helpful and they will make this paper stand out as a go-to guide for researchers at all levels.

I have several recommendations to further reinforce the quality of this manuscript and to enhance its accessibility to wider audiences, such as those from different fields or non-native English speakers. Most of my suggestions aim to improve clarity through added definitions and examples.

1. I recommend changing "empirical psychology" to "quantitative psychology" throughout the paper, as empirical research is not synonymous with quantitative research.
2. Page 3: "recuding" Typo should be corrected.
3. Page 4: "during manuscript preparation or as a benchmark for evaluating the reporting quality of existing studies." This part would benefit from including teaching as an additional use case.
4. Page 5: "Post-hoc analyses, including deviations from preregistered plans or exploratory analyses, are often necessary" It might be helpful to add that this is normal and expected. Many researchers are hesitant to preregister because they worry about mistakes made during the planning phase, only noticed after data collection or during analysis.
5. Page 5: "Lastly, researchers can also split datasets into two parts..." Please clarify whether this refers to situations where data have already been collected before preregistration. In addition, emphasizing that preregistration can still occur after data collection (especially with pre-existing data) would be valuable.
6. Page 5: "Ultimately, while preregistration can serve as a valuable mechanism framework, for enhancing transparency and rigor within an 'error statistical' it should not be regarded as universally necessary, data particularly when it risks favoring a single arbitrary perspective over a more comprehensive examination of the data." I appreciated this sentence. I applaud the authors for including this important nuance.
7. Page 5: Sample size planning Consider discussing sensitivity analysis for cases where data have already been collected. Signposting the section at the end of the paper that discourages post hoc power analysis would also be very useful here.
8. Page 6: "for example in replication studies), the desired significance level (usually set to $\alpha = 0.05$), and the effect size they anticipate to observe." For clarity, it would help to note or signpost where researchers can obtain an expected effect size, especially as the next section addresses this.
9. What is the question mark reference? I noticed this symbol throughout the paper and it needs clarification.
10. Although the text mentions that packages are for R, adding this detail explicitly in the tables as done in Table 4 (e.g., "R: package name") would improve clarity and consistency.
11. Page 9: "This information entails the content of the experimental manipulation and the method, time and location of manipulation delivery." I suggest also adding the need to describe all experimental and control conditions here.
12. Page 9: To improve flow, the paragraph beginning with "In some cases..." would be better placed after the paragraph starting with "Outliers refer...", and before the discussion on missing data begins.
13. Page 10: "adhering to these guidelines, researchers can ensure the robustness and reproducibility of their findings." I recommend adding that this also enables accurate interpretation of results, which is arguably the most critical issue.
14. Page 11: Figure 1 For consistency with the text, consider adding a "scatter plot" after "bar plot."
15. Page 17: "Therefore, it is recommended to use bias-corrected effect size estimates..." Table 5 is referred to earlier without context; it would be helpful to refer back to this table again after this sentence.
16. Page 17: Table 5 Even though earlier tables mention recommended software packages, adding a dedicated column in Table 5 for software would help readers easily find tools for calculating the listed effect sizes.
17. Page 17: "Post-hoc power analysis ("observed power")" Please clarify whether sensitivity analysis is the same as post-hoc power analysis. This distinction is often misunderstood and better guidance would be very helpful, especially as reviewers frequently request such analyses.
18. To aid readability and support learning, I suggest bolding key terms throughout the paper (e.g., file drawer problem,

publication bias, Mayo's error statistical framework).

19. To improve accessibility, I recommend briefly defining the following terms:

- Page 4: Mayo's error statistical framework; severe hypothesis testing
- Page 8: beta-spending function; alpha-spending function
- Page 9: manipulation check
- Page 9: complete case analysis (listwise deletion); multiple imputation; full information maximum likelihood; robust estimators; bootstrapping
- Page 10: citation reports
- Page 13: Neyman-Pearson framework

20. The example sentences for reporting (including both good and poor reporting practices) are incredibly helpful. I recommend adding examples for the following areas as well:

- Page 4: "If a construct can be measured in multiple ways, researchers should acknowledge these variations and justify their chosen approach."
- Page 4: "Hypotheses that are too narrow risk becoming trivial, whereas those that are too broad may be unfalsifiable." Examples for too narrow and too broad hypotheses
- Page 6: "An increasingly recommended alternative is to define the minimal effect size of interest based on theoretical or practical considerations."
- Page 7: "Generally, it is advisable to base the sample size on the test that requires the largest sample to ensure adequate power across all analyses."
- Page 12: "Reporting results of Bayesian methods" Examples of good vs bad reporting
- Page 12: "Reporting results of sequential designs" Examples of good vs bad reporting
- Page 13: "Proper interpretation requires understanding that a lack of statistical significance does not equate to the absence of an effect. It may instead reflect data insensitivity." Examples of good vs bad reporting
- Page 14: "Regardless of the test framework used, all relevant aspects of the sequential design need be transparently reported, ideally accompanied by preregistration, to ensure the evidence presented is compelling and credible." Examples of good vs bad reporting
- Page 15: "In cases where the software does not offer correction for conditional bias, reports should explicitly acknowledge this limitation."
- Page 17: "Therefore, pre-specifying the correction approach and number of corrections can increase the falsifiability of a test."

Reviewer #2 (Remarks to the Author):

The stated goal of the current study is to "offer[s] a structured set of guidelines for reporting statistical analyses in empirical psychology, emphasizing clarity at both the planning and results stages". There is much to like about this paper. The TSRP checklist is a great tool that I could see being widely used. I could also see the paper itself being used in advanced graduate student training curricula. However, the split focus between fixed-sample designs and sequential designs is very ambitious, and at times it feels like depth is being sacrificed for breadth. My comments below are mostly minor and geared toward enhancing an already strong paper.

It is unclear if the authors are intending to use NHST and frequentist paradigms interchangeably.

It would be helpful to add citations for the description of Bayesian statistics in Box 1.

Please spell out SHAP (Box 3) at first use.

The description of tables on page 3 skips Table 3.

It is not clear what is meant by "imprecise, ad hoc verbal hypotheses" (p. 4)

Please spell out ASN in the header and describe it at first use (it appears in the table on p. 8 before appearing in the main text)

There appears to be a missing citation (a question mark appears instead) on page 12.

Please spell out SBF at first use (p. 12)

TOST is not spelled out (p. 13) or described.

It may be more straightforward to label Box 3 as just Box 3, rather than Figure 2 and Box 3.

Table 6 is excellent and may be useful to include earlier in the paper. The cognitive psychology aspect of the example tends to get lost in the text. It would also be nice to have additional best practice guidance on describing the result of the Condition x Time interaction (i.e., which condition increased/decreased more/less over time?).

Reviewer #3 (Remarks to the Author):

Improving Statistical Reporting in Psychology

by Anna-Lena Schubert , Meike Steinhilber , Heemin Kang , Daniel S. Quintana

This is an ambitious paper that aims to provide comprehensive recommendations for improving reporting practices in psychological research. The paper presents many valuable points and compiles a wide range of references that will be useful for many researchers. However, I have some reservations, particularly regarding the framing of statistical inference frameworks, which should be addressed before the paper is suitable for publication. Below are some comments, roughly following the order of the paper, though not necessarily reflecting their importance.

Major comments

1) It could perhaps be more illustrative to use a real and not a hypothetical study.

2) The presentation of "Null Hypothesis Significance Testing" (NHST) as a well-accepted statistical framework is misleading. In reality, NHST is a confused blend of Fisherian and Neyman-Pearsonian hypothesis testing approaches that is used in practice, but neither Fisher nor Neyman-Pearson advocated it. While Box 1 briefly mentions this, the key conceptual differences could be more stressed:

- In Fisherian testing there is only a null hypothesis H_0 , p-values are interpreted as quantitative evidence against H_0 , H_0 can never be accepted, only rejected or not rejected.

- In Neyman-Pearson testing, there is a null hypothesis H_0 and an alternative hypothesis H_1 , p-values are only a byproduct of a binary decision procedure aimed at controlling the long-run error rates α and β , both H_0 or H_1 can be accepted. Here are some useful references: Hubbard and Bayarri (2003, 10.1198/0003130031856), Goodman (1993, 10.1093/oxfordjournals.aje.a116700), Goodman (2016, 10.1126/science.aaf5406), Greenland (2019, 10.1080/00031305.2018.1529625), Greenland (2023, 10.1111/sjos.12625), and Gigerenzer (2004, 10.1016/j.socec.2004.09.033), though there are many more.

To avoid reinforcing common misconceptions, these distinctions should be made explicit throughout the paper. For example, the sentence

"The statistical power of a test reflects the probability of correctly rejecting the null hypothesis if it is indeed false (which practically means correctly accepting the alternative hypothesis)"

conflates Fisherian and Neyman-Pearsonian concepts, contributing to conceptual confusion.

3) The distinction between exploratory and confirmatory research is crucial and should be introduced earlier. The paper appears to focus primarily on confirmatory research, yet in practice, much empirical research is exploratory.

4) The "Bayesian approach" discussed in the paper seems almost entirely focused on Bayes factor hypothesis testing. For example, the sentence

"Bayesian inference, in contrast, evaluates the relative plausibility of hypotheses given the data"

suggests this. However, Bayes factor hypothesis testing is only one flavor of Bayesian statistics. In practice, researchers may also use other approaches, such as Bayesian estimation or posterior probabilities.

5) "Sequential designs" is positioned as an alternative paradigm to Bayesian and frequentist inference, which seems misleading. Frequentist and Bayesian statistics differ in terms of the notion of probability employed (aleatory vs. epistemic). Sequential designs are unrelated to notions of probability but relate to study design. Rather, the counterpart of sequential designs are fixed designs. Both types of designs can be used in frequentist and Bayesian statistics.

6) Equivalence testing is positioned as a post hoc step following non-significant results, e.g.,

"In contrast, if the data does not significantly deviate from what is expected under H_0 , additional equivalence testing should be conducted before concluding the absence of a theoretically or practically relevant effect or difference"

perhaps similar to the "Conditional equivalence testing" approach from Campell and Gustafson (2018, 10.1371/journal.pone.0195145). However, equivalence testing, as originally intended, is a procedure to test the null hypotheses that a parameter is outside an equivalence range (as shown in your Box 3) and this kind of question arises naturally in many studies, e.g., when attempting to "prove" that a new and cheaper treatment is equally good as an established but more expensive treatment, hence the term "equivalence". It could be useful to clarify this. Finally, the sample size required for achieving satisfactory power in an equivalence test is typically much higher than for a point null hypothesis test, so when following the advocated post hoc equivalence testing approach, the equivalence test will often be underpowered even if the true effect is actually negligible.

Minor comments

1) The term "poor reporting" feels somewhat judgmental. A more constructive alternative might be "incomplete" or "insufficient".

2) Introduction "Given that seeking additional details or clarifications from authors can be challenging and unreliable": Can this statement be substantiated with some references?

3) Paragraph "Multiple statistical tests": The section on multiple tests should mention how multiplicity adjustments (e.g., Bonferroni correction) can be incorporated at the design stage to adjust for multiplicity.

4) Box 2

- "exploiting the fact that p-values do not converge under the null hypothesis"

In this statement, it was not clear to me to "what" p-values should converge under the null? Do the authors perhaps mean that p-values are uniformly distributed under the null? Also, I don't think anything is "exploited".

- "On average, an SPRT requires 50% fewer data points than a test using a fixed-sample design"

Over what is the "average" taken? I have doubts that such a general statement can be made as usually such operating characteristics depend on the exact setting (distribution of the data, effect sizes, number of interim analyses, etc.). This sentence needs to be further clarified or toned down.

- It could be additionally mentioned that sequential designs can be logistically challenging (e.g., the actual costs of the study are not known in advance, interim analyses require unblinding).

5) Table 1: Another useful reference for blinded analyses is Dutilh (2021, 10.1007/s11229-019-02456-7)

6) "Bayesian frameworks: Precision and simulation": The section suggests that Bayesian sample size planning always involves simulation, which is not the case. See, e.g., the articles by Wong and Tendeiro (2025, 10.3758/s13428-025-02654-x), Pawel and Held (2025, 10.1080/00031305.2025.2467919), and Pawel et al. (2023, 10.1037/met0000604), which could be mentioned.

7) Table 3 could also mention the BFDA package for the SBFT (<<https://github.com/nicebread/bfda>>)

8) The acronym "ASN" is used before being formally introduced, this should be corrected.

9) "This choice depends on the type of data collected, the assumptions of different statistical methods, whether the study follows a frequentist, Bayesian, or sequential approach.": This phrase suggests that these approaches are mutually exclusive. However, in practice, a study may combine approaches (e.g., report both Bayesian and frequentist analyses), which can enhance robustness.

10) While the grateful package is useful for generating citation reports, it may be worth mentioning that simple citations can be obtained using `citation(package = "packagename")` in base R.

11) "Alternative methods, such as the scatterplot, raincloud plot and violin plot can be more informative to examine data distributions": Maybe boxplots could also be mentioned as they improve upon barplots?

12) "Exact p-values can be informative for readers, and can aid future meta-scientific work, such as calculating effect sizes for meta-analysis when other statistics are not available.": While exact p-values facilitate meta-analysis, their most fundamental function is providing a quantitative measure of evidence against the null hypothesis, which should be mentioned.

14) "While NHST remains prevalent, its conventional application has well-known limitations, including its reliance on arbitrary significance thresholds and sensitivity to sample size.": Many of these criticisms stem from misunderstandings of p-values and the conflation of Fisherian and Neyman-Pearsonian ideas, see e.g., Greenland (2019, 10.1080/00031305.2018.1529625). Furthermore, Bayes factor thresholds also rely on arbitrary conventions, so this critique should be presented with more nuance.

15) Equivalence testing: It would be important to say that not just any confidence interval can be used but one should use an $(1 - 2\alpha)100\%$ CI to obtain an equivalence test with type-I error rate at most α (e.g., a 90% CI for a 5% level). Equivalently, one could do two one-sided tests at the margin (the TOST procedure) instead of the CI approach, which could also be mentioned.

16) "it is essential to pre-specify equivalence bounds before data analysis to avoid bias": The authors advocated earlier that equivalence tests should be run post hoc when observing null results, hence in most situations researcher would probably not prespecify the margin as they are expecting a genuine effect a priori. Or would the authors suggest to also specify a margin, even when first performing a point null hypothesis test? A useful reference that discusses such post hoc specification issues may be Campbell and Gustafson (2023, 10.15626/MP.2020.2506)

17) Table 4: In my view, all regression models in the Table should also be associated with regression coefficients as effect sizes. Usually, one fits a regression model to "adjust" the effect of a treatment of interest with respect to other variables. In this situations, the estimated treatment coefficient (or a transformation, e.g., exp in logistic regression to obtain an odds ratio) is the effect size of interest and not R^2 . Moreover, some of these effect sizes seem more like goodness of fit measures or information criteria (e.g., R^2 or BIC) rather than actually quantifying the size of an effect in an interpretable way.

18) "Misinterpretation of p-values": I would say that "trending towards significance" is a minor misinterpretation compared to many more severe misinterpretations and misuses of p-values, e.g., interpreting the p-value as the posterior probability of H_0 , or as the probability that "chance" generated the data. I recommend adding more severe ones, see e.g., Greenland et al. (2016, 10.1007/s10654-016-0149-3).

19) Paragraph on post-hoc power: The paper by Hoenig and Heisey (2001, 10.1198/000313001300339897) could be cited.

Version 1:

Decision Letter:

** Please ensure you delete the link to your author homepage in this e-mail if you wish to forward it to your co-authors **

Dear Professor Schubert, Dear Anna-Lena,

Your Perspective titled "Improving Statistical Reporting in Psychology" has now been editorially reviewed, and I am delighted to say that we are happy, in principle, to publish it in Communications Psychology.

We will not send your revised paper for further review if, in the editors' judgement, the revised paper is in Communications Psychology format, in accessible style and of appropriate length, we shall accept it for publication immediately. I have attached an edited version of your manuscript and an editorial request table, and ask you to attend to each comment in detail.

EDITORIAL REQUESTS:

* Please review the changes in the attached copy of your manuscript, which has been edited for style, and address the comments and queries I have added. If using Word, please use the 'track changes' feature to make the process of accepting your manuscript more efficient.

* Communications Psychology uses a transparent peer review system. On author request, confidential information and data can be removed from the published reviewer reports and rebuttal letters prior to publication. If you are concerned about the release of confidential data, please let us know specifically what information you would like to have removed. Please note that we cannot incorporate redactions for any other reasons.

*If you have not done so already, please alert me to any related manuscripts from your group that are under consideration or in press at other journals, or are being written up for submission to other journals (see www.nature.com/authors/editorial_policies/duplicate.html for details).

SUBMISSION INFORMATION:

* If you wish, you may also submit a visually arresting image, together with a concise legend, for consideration as a 'Hero Image' on our homepage. The file should be 1400x400 pixels and should be uploaded as 'Related Manuscript File'. In addition to our home page, we may also use this image (with credit) in other journal-specific promotional material.

In order to accept your paper, we require the following:

* The final version of your text as a Word or TeX/LaTeX file, with any tables prepared using the Table menu in Word or the table environment in TeX/LaTeX and using the 'track changes' feature in Word.

* Production-quality versions of all figures, supplied as separate files. Photographic images should be 300 dpi in RGB format (.jpg, TIFF or native Photoshop format) and any labels/scale bars included in a separate layer from the image. Line art, graphs and schemes should be vector format (.ai, .eps, .pdf); Adobe Illustrator files are preferred and will minimize production time. Any chemical structures or schemes contained within figures should additionally be supplied as separate Chemdraw (.cdx) files.

Communications Psychology is a fully open access journal. Articles are made freely accessible on publication. For further information about article processing charges, open access funding, and advice and support from Nature Research, please visit <https://www.nature.com/commpsychol/open-access>

At acceptance, you will be provided with instructions for completing the open access licence agreement on behalf of all authors. This grants us the necessary permissions to publish your paper. Additionally, you will be asked to declare that all required third party permissions have been obtained.

Please note that your paper cannot be sent for typesetting to our production team until we have received this information; **therefore, please ensure that you have this ready when submitting the final version of your manuscript.**

ORCID

Communications Psychology is committed to improving transparency in authorship. As part of our efforts in this direction, we are now requesting that all authors identified as 'corresponding author' create and link their Open Researcher and Contributor Identifier (ORCID) with their account on the Manuscript Tracking System (MTS) prior to acceptance. ORCID helps the scientific community achieve unambiguous attribution of all scholarly contributions. For more information please visit <http://www.springernature.com/orcid>

For all corresponding authors listed on the manuscript, please follow the instructions in the link below to link your ORCID to your account on our MTS before submitting the final version of the manuscript. If you do not yet have an ORCID you will be able to create one in minutes.

IMPORTANT: All authors identified as 'corresponding author' on the manuscript must follow these instructions. Non-corresponding authors do not have to link their ORCIDs but are encouraged to do so. Please note that it will not be possible to add/modify ORCIDs at proof. Thus, if they wish to have their ORCID added to the paper they must also follow the above procedure prior to acceptance.

To support ORCID's aims, we only allow a single ORCID identifier to be attached to one account. If you have any issues attaching an ORCID identifier to your MTS account, please contact the [Platform Support Helpdesk](http://platformsupport.nature.com/).

Link Redacted

We hope to hear from you within two weeks; please let us know if the process may take longer.

Best regards,

Jennifer

Jennifer Bellingtier, PhD
Senior Editor
Communications Psychology

Reviewer #1:

This paper offers clear and constructive guidance on statistical reporting in psychology. It will be very useful for researchers, reviewers, and editors, but perhaps most importantly for students and early-career researchers who are trying to grasp the evolving norms of a rapidly improving field. I also expect this paper to be used in teaching, in addition to informing research practices.

I enjoyed reading it and appreciate its clarity and thoughtful signposting to additional resources where appropriate. The example sentences provided are especially helpful and they will make this paper stand out as a go-to guide for researchers at all levels.

I have several recommendations to further reinforce the quality of this manuscript and to enhance its accessibility to wider audiences, such as those from different fields or non-native English speakers. Most of my suggestions aim to improve clarity through added definitions and examples.

We thank the reviewer for their generous and encouraging feedback. We're pleased that the paper is seen as a helpful resource for researchers at all levels, and that its clarity, examples, and signposting were appreciated. We also value the suggestions for improving accessibility and clarity and have carefully considered them in our revisions.

(1) I recommend changing “empirical psychology” to “quantitative psychology” throughout the paper, as empirical research is not synonymous with quantitative research.

We appreciate this suggestion and changed the terms and related terms throughout the revised manuscript.

(2) Page 3: “recuding” ◇ Typo should be corrected.

We fixed the typo.

(3) Page 4: “during manuscript preparation or as a benchmark for evaluating the reporting quality of existing studies.” ◇ This part would benefit from including teaching as an additional use case.

This is a great suggestion – we added teaching as a use case on p. 2: “Readers can use these examples as guidance during manuscript preparation, for teaching, or as a benchmark for evaluating the reporting quality of existing studies.”

(4) Page 5: “Post-hoc analyses, including deviations from preregistered plans or exploratory analyses, are often necessary” ◇ It might be helpful to add that this is normal and expected. Many researchers are hesitant to preregister because they worry about mistakes made during the planning phase, only noticed after data collection or during analysis.

We agree this it is helpful to normalize such deviations and edited the section on p. 6 accordingly (changes highlighted): “Post-hoc analyses, including deviations from preregistered plans or exploratory analyses, are often necessary—for instance, when statistical models fail to converge or distributional assumptions are violated⁷⁸. Such deviations are normal and expected, as oversights in the planning phase or analytical challenges are often only revealed during data collection or analysis. Transparent documentation of such deviations allows readers to assess their impact, which can promote credibility and methodological rigor^{71,72}. “

(5) Page 5: “Lastly, researchers can also split datasets into two parts...” ◇ Please clarify whether this refers to situations where data have already been collected before preregistration. In addition, emphasizing that preregistration can still occur after data collection (especially with pre-existing data) would be valuable.

We thank the reviewer for this helpful suggestion. In response, we clarified that preregistration can still be valuable when data have already been collected, particularly with pre-existing datasets, on p. 6 (changes highlighted): “A time-stamped, pre-specified analysis plan further constrains researcher flexibility, reducing opportunities for post-hoc adjustments that could inflate false positives. Importantly, preregistration can still be useful even when data have already been collected, such as when working with pre-existing datasets⁶⁶.”

In addition, we clarified that splitting datasets is a strategy particularly helpful when working with pre-existing datasets on p. 6 (changes highlighted): “Lastly, researchers can also split pre-existing datasets into two parts, using the first part to assess distributional properties and modeling assumptions and then incrementally preregistering their analysis plans for the second part⁷⁵.”

(6) Page 5: “Ultimately, while preregistration can serve as a valuable mechanism framework, for enhancing transparency and rigor within an ‘error statistical’ it should not be regarded as universally necessary, particularly when it risks favoring a single arbitrary perspective over a more comprehensive examination of the data.” ◇ I appreciated this sentence. I applaud the authors for including this important nuance.

We appreciate the reviewer’s positive feedback on this point. We agree that it is important to emphasize the value of preregistration without presenting it as a universal requirement, especially when flexibility may lead to a more comprehensive understanding of the data.

(7) Page 5: Sample size planning ◇ Consider discussing sensitivity analysis for cases where data have already been collected. Signposting the section at the end of the paper that discourages post hoc power analysis would also be very useful here.

We agree that discussing sensitivity analysis is an important addition and added a brief discussion of it on p. 6: “In cases where data have already been collected, *sensitivity analysis* offers a meaningful alternative to traditional power analysis. Rather than estimating the required sample size, sensitivity analysis identifies the smallest effect size that the study is capable of reliably detecting given the available sample, chosen alpha level, and desired power⁸⁰. This approach is particularly useful for contextualizing null results or assessing the robustness of observed effects. Notably, this is preferable to post hoc power analysis, which is widely discouraged due to its misleading interpretation of non-significant results⁸¹ (see Section “Common errors and misinterpretations” below).”

(8) Page 6: “for example in replication studies), the desired significance level (usually set to $\alpha = 0.05$), and the effect size they anticipate to observe.” \diamond For clarity, it would help to note or signpost where researchers can obtain an expected effect size, especially as the next section addresses this.

We thank the reviewer for pointing this out. We added a brief forward reference to the subsequent section, which addresses common issues in obtaining expected effect sizes for power analysis, on p. 7 (changes highlighted): “When determining the required sample size, researchers have to specify the specific statistical test they intend to use (e.g., a bivariate correlation, a within-subjects t-test, or a 3-way ANOVA), the statistical power they aim to achieve (usually at least $1 - \beta = 0.80$, but a higher power can be desirable if researchers want to be more confident about their findings—for example in replication studies), the desired significance level (usually set to $\alpha = 0.05$), and the effect size they anticipate to observe (see next section for a discussion of where expected effect sizes can be obtained and common pitfalls in using them).”

(9) What is the question mark reference? I noticed this symbol throughout the paper and it needs clarification.

This was an error in typesetting – the missing reference is a paper by Van De Schoot and colleagues (2021), which we added back in when revising the manuscript.

(10) Although the text mentions that packages are for R, adding this detail explicitly in the tables as done in Table 4 (e.g., “R: package name”) would improve clarity and consistency.

Thank you for this suggestion. We agree that this is helpful but ultimately decided against adding it before each package (because it created a lot of redundancy in tables with a large number of R packages) and instead used the column title “R packages” whenever we refer to software packages in tables.

In addition, we decided to remove the few generic references to other software (SPSS, JASP, Mplus) from Table 5.

(11) Page 9: “This information entails the content of the experimental manipulation and the method, time and location of manipulation delivery.” ◇ I suggest also adding the need to describe all experimental and control conditions here.

We modified the sentence on p. 9 as follows (changes highlighted):” This information entails the content of the experimental manipulation and the method, time and location of manipulation delivery, including a description of all experimental and control conditions.”

(12) Page 9: To improve flow, the paragraph beginning with “In some cases...” would be better placed after the paragraph starting with “Outliers refer...”, and before the discussion on missing data begins.

We appreciate this suggestion and restructured this section as suggested by the reviewer.

(13) Page 10: “adhering to these guidelines, researchers can ensure the robustness and reproducibility of their findings.” ◇ I recommend adding that this also enables accurate interpretation of results, which is arguably the most critical issue.

This is a great suggestion, which we added on p. 12 (changes highlighted): “By adhering to these guidelines, researchers can ensure the robustness, reproducibility, and accurate interpretation of their findings.”

(14) Page 11: Figure 1 ◇ For consistency with the text, consider adding a “scatter plot” after “bar plot.”

We agree that consistency across the text and figures is important. We therefore now mention boxplots instead of scatterplots as examples of better figures on p. 11 (changes highlighted): “However, it has repeatedly shown that barplots may make it difficult for readers to see the data distribution ^{148,149}. Alternative methods, such as the boxplot ¹⁴⁹, raincloud plot ¹⁴⁸ and violin plot ¹⁵⁰ can be more informative to examine data distributions, as illustrated in Figure 1.”

(15) Page 17: “Therefore, it is recommended to use bias-corrected effect size estimates...” ◇ Table 5 is referred to earlier without context; it would be helpful to refer back to this table again after this sentence.

This is a helpful suggestion: We added a reference to this table (now Table 6) at the end of this sentence.

(16) Page 17: Table 5 ◇ Even though earlier tables mention recommended software packages, adding a dedicated column in Table 5 for software would help readers easily find tools for calculating the listed effect sizes.

We appreciate this suggestion and added a dedicated software column including a reference to the effectsize package to the table (now Table 6).

(17) Page 17: “Post-hoc power analysis (“observed power”)” ♦ **Please clarify whether sensitivity analysis is the same as post-hoc power analysis. This distinction is often misunderstood and better guidance would be very helpful, especially as reviewers frequently request such analyses.**

We thank the reviewer for highlighting this point. To address this common misconception, we have added a clarifying note distinguishing sensitivity analysis from post-hoc power analysis, emphasizing why the former is a more informative and statistically appropriate approach. This clarification now appears in the section on post-hoc power analysis on p. 18 (changes highlighted): “Rather than relying on post-hoc power calculations, researchers should focus on reporting the precision of their estimates through confidence intervals around effect sizes²⁴⁷. In cases of non-significant findings, equivalence testing or Bayesian approaches can provide more informative interpretations regarding the presence or absence of meaningful effects. Another useful approach is sensitivity analysis, which is often mistakenly conflated with post-hoc power analysis. However, the two differ conceptually and inferentially: while post-hoc power analysis uses the observed effect size and thus reiterates the p -value⁸¹, sensitivity analysis estimates the smallest effect size that a study could have detected with reasonable power, given its sample size and alpha level²⁴⁸. This makes sensitivity analysis particularly valuable when interpreting null results from secondary or archival data. Ultimately, the validity of statistical inferences should rest on well-documented a priori power analyses or transparent alternatives like sensitivity analysis—not retrospective justifications that conflate statistical power with observed outcomes.”

(18) To aid readability and support learning, I suggest bolding key terms throughout the paper (e.g., file drawer problem, publication bias, Mayo’s error statistical framework).

We appreciate the suggestion to bold key terms. To preserve typographic consistency, we prefer not to bold terms throughout the manuscript unless instructed otherwise by the Editor. Instead, we (i) define each key term at first use in the text and (ii) provide a companion glossary in the supplementary materials that consolidates brief definitions for easy reference (see <https://osf.io/xtq6s/>). This approach supports readability and learning while keeping the running text accessible.

(19) To improve accessibility, I recommend briefly defining the following terms:

- **Page 4: Mayo’s error statistical framework; severe hypothesis testing**
- **Page 8: beta-spending function; alpha-spending function**
- **Page 9: manipulation check**

- Page 9: complete case analysis (listwise deletion); multiple imputation; full information maximum likelihood; robust estimators; bootstrapping

- Page 10: citation reports

- Page 13: Neyman-Pearson framework

We appreciate the reviewer’s suggestion to define technical terms for broader accessibility. We have added brief in-text definitions for the highlighted terms at first mention, ensuring clarity for readers who may be less familiar with these concepts.

pp. 5-6: “Within Mayo’s *error statistical* framework, which focuses on how statistical methods can expose potential errors in hypothesis testing, preregistration allows for more severe hypothesis testing: the more specific and falsifiable a preregistered claim (e.g., a predicted effect size range), the stronger its inferential value.”

p. 9: “In general, GSDs retain a maximum sample size and pre-specified number of interim looks, with control over the Type I error rate via alpha-spending functions (which control the cumulative Type I error rate across interim analyses) and over the Type II error rate with optional beta-spending functions (which manage the Type II error rate).”

p. 9: “It is also essential to include a manipulation check—an empirical test to verify that an experimental manipulation successfully influenced the intended construct—to ensure that the manipulation targets the construct as intended.”

p. 10: “Complete case analysis (also known as list-wise deletion) is the most commonly reported missing data treatment method^{138,139}.”

p. 10: “Instead, multiple imputation^{136,141}, a method that replaces missing values with a set of plausible values based on observed data, or full information maximum likelihood¹⁴², which estimates model parameters directly using all available data without imputing missing values, are recommended to address missing data^{139,143} (but see¹⁴⁴ for a comparison of the two methods).”

pp. 9-10: “Robust estimators, such as M-estimators or trimmed means, which reduce the influence of extreme values on parameter estimates, and nonparametric resampling techniques such as bootstrapping, a resampling method that repeatedly draws samples from the observed data to estimate the sampling distribution, offer alternatives that can accommodate outliers without violating model assumptions or inflating Type I error rates¹³⁴.”

p. 10: “The grateful R package can be used for generating citation reports, which automatically compile references for the R packages used in an analysis¹⁴⁵.”

p. 13: “Within the Neyman-Pearson framework (see Box 1), where both null and alternative hypotheses are explicitly defined and the study is sufficiently powered, failing to reject the null leads to a decision to act as if the null were true. Crucially,

this decision is made with the recognition that it may be incorrect in any single instance, though error rates are controlled over repeated sampling.”

(20) The example sentences for reporting (including both good and poor reporting practices) are incredibly helpful. I recommend adding examples for the following areas as well.

We appreciate this suggestion and added further examples to the highlighted areas:

p. 5: If a construct can be measured in multiple ways, researchers should acknowledge these variations and justify their chosen approach. For example, instead of the vague statement, “Working memory capacity was assessed using a digit span task,” a more transparent description would be: “Working memory capacity was assessed using a backward digit span task. While several tasks could be used to assess this construct (e.g., complex span, n-back), we selected this task due to its widespread use and strong psychometric properties in prior visual-verbal memory research.”

p. 5: Hypotheses that are too narrow risk becoming trivial, whereas those that are too broad may be unfalsifiable. For instance, “We hypothesize that cognitive control influences behavior” is overly broad and unfalsifiable, whereas “We expect a 35-ms difference in RT between conditions on trial 3” is overly narrow and likely trivial. A more balanced hypothesis might be: “We hypothesize that participants with higher working memory capacity will show reduced switch costs (as indexed by reaction time differences) in a task-switching paradigm compared to participants with lower capacity.” Researchers should strive for precision while ensuring their hypotheses remain relevant and applicable to the broader theoretical context.”

p. 7: An increasingly recommended alternative is to define the minimal effect size of interest based on theoretical or practical considerations. For example, instead of stating, “We based our sample size on an expected medium effect size,” a more informative approach is: “We defined a minimal effect size of interest as $d = 0.30$, based on prior meta-analytic estimates and the smallest effect that would have theoretical and applied relevance for the cognitive training intervention.”

p. 8: Generally, it is advisable to base the sample size on the test that requires the largest sample to ensure adequate power across all analyses. For example, rather than stating, “Sample size was determined based on the main ANOVA,” one could write: “Sample size was based on the mediation analysis, which required a larger sample than the primary group comparison, ensuring sufficient power across all planned analyses.”

p. 12: The computed Bayes factors and the method used to calculate them (e.g., default JZS prior, Savage-Dickey ratio, or bridge sampling) should be reported, including the software and version (e.g., JASP, BayesFactor, bridgesampling). For

example, instead of stating, “We found a Bayes factor of 3”, a more informative report would be: “Using a default JZS prior, we found moderate evidence for the alternative hypothesis ($BF_{10} = 3.21$), indicating that the data were approximately three times more likely under H_1 than under H_0 . All priors and model specifications are provided in the Supplementary Materials.”

p. 13: When reporting results from sequential methods, researchers should include the type of test used, decision boundaries, planned stopping points, final sample size, and any adjustments made during data collection. For instance, vague reporting such as “Data collection stopped once a significant result was observed” should be replaced with: “We used a sequential Bayes factor design with stopping thresholds of $BF_{10} > 6$ and $BF_{01} > 6$, computed every 10 participants. Data collection stopped at $N = 80$ when $BF_{10} = 7.2$. Sensitivity and robustness checks are reported in the Supplementary Materials.”

pp. 13-14: Proper interpretation requires understanding that a lack of statistical significance does not equate to the absence of an effect. It may instead reflect data insensitivity. For example, rather than writing, “The effect was not significant, so we conclude there is no effect,” it is more accurate to state: “The effect was non-significant ($p = .12$), but the 95% CI $[-0.10, 0.35]$ includes values that could still be of practical importance. Thus, the data were inconclusive, and further research with increased statistical power is warranted.”

p. 15: Regardless of the test framework used, all relevant aspects of the sequential design need be transparently reported, ideally accompanied by preregistration, to ensure the evidence presented is compelling and credible. For example, “We used a sequential design and stopped at $N = 60$ ” is insufficient. Instead, one might report: “We preregistered a group sequential design with three interim looks, using an O’Brien-Fleming alpha-spending function to maintain the familywise Type I error rate at .05. The study was stopped at the second interim analysis ($N = 60$) after reaching the critical boundary.”

p. 16: In cases where the software does not offer correction for conditional bias, reports should explicitly acknowledge this limitation. In cases where the software does offer correction for conditional bias, reports should explicitly acknowledge this limitation. For example: “The estimated effect size was $d = 0.56$. However, this value may be upwardly biased due to early stopping in the sequential design. Because our software did not implement conditional bias correction, we encourage caution in interpreting the magnitude of this estimate.”

p. 18: Therefore, pre-specifying the correction approach and number of corrections can increase the falsifiability of a test. For example: “We preregistered that Holm-Bonferroni correction would be used across four primary comparisons to control the familywise error rate at $\alpha = .05$.”

Reviewer #2:

The stated goal of the current study is to “offer[s] a structured set of guidelines for reporting statistical analyses in empirical psychology, emphasizing clarity at both the planning and results stages”. There is much to like about this paper. The TSRP checklist is a great tool that I could see being widely used. I could also see the paper itself being used in advanced graduate student training curricula. However, the split focus between fixed-sample designs and sequential designs is very ambitious, and at times it feels like depth is being sacrificed for breadth. My comments below are mostly minor and geared toward enhancing an already strong paper.

We sincerely thank the reviewer for their thoughtful and constructive comments. We are especially grateful for the recognition of the TSRP checklist and the paper’s potential utility in graduate training contexts. Your feedback has been invaluable in helping us refine and clarify the manuscript. We appreciate your acknowledgment that our dual focus on fixed-sample and sequential designs is ambitious, and we have made targeted revisions to improve clarity and depth where feasible. Most of your suggestions were straightforward to address, and we have incorporated each of them to enhance the accessibility, precision, and instructional value of the paper. Below, we respond to each point in detail.

(1) It is unclear if the authors are intending to use NHST and frequentist paradigms interchangeably.

Thank you for the comment, we now clarify this in Box 1 (changes highlighted):

Frequentist statistics encompasses two distinct statistical theories that have historically been intermingled and often confused, despite their fundamental and incompatible differences. For further context, see ⁷⁻¹¹, as well as the tutorial by ¹² for pedagogical guidance. The term null hypothesis significance testing (NHST) refers to a hybrid of these two frameworks and is therefore deliberately avoided in this paper.

(2) It would be helpful to add citations for the description of Bayesian statistics in Box 1.

We would like to thank the reviewer for this helpful suggestion and added two references to Box 1:

Etz, A. & Vandekerckhove, J. Introduction to Bayesian inference for psychology. *Psychon Bull Rev* **25**, 5–34 (2018).

van de Schoot, R. *et al.* Bayesian statistics and modelling. *Nat Rev Methods Primers* **1**, 1–26 (2021).

(3) Please spell out SHAP (Box 3) at first use.

We now spell out SHAP at its first use (changes highlighted): “As a complementary resource, see ³⁹ for an introduction to SHAP (SHapley Additive exPlanations) values, a general model-agnostic approach for interpreting how individual input features contribute to model predictions.”

(4) The description of tables on page 3 skips Table 3.

We thank the reviewer for catching this oversight. We have revised the sentence on page 3 to include all tables, ensuring that all key resources are appropriately referenced and the list of supporting materials is complete.

(5) It is not clear what is meant by “imprecise, ad hoc verbal hypotheses” (p. 4)

We appreciate the reviewer’s request for clarification. We have revised the relevant sentence (now on p. 5) to include a brief explanation and example of what constitutes an “imprecise, ad hoc verbal hypothesis” (changes highlighted):

“Precise and operationalizable hypotheses are essential for theory-testing confirmatory research. However, psychological theories are often ambiguous, which hinders cumulative knowledge building⁵⁶. A major issue is the reliance on imprecise, ad hoc verbal hypotheses. These often lack specificity regarding the variables involved, the expected direction or magnitude of effects, and the underlying mechanisms, thereby allowing excessive interpretative flexibility.

For instance, a researcher might hypothesize that “stress impairs memory performance”. While seemingly intuitive, this claim is underspecified: it does not define what kind of stress is being studied (e.g., acute vs. chronic, physiological vs. perceived), how memory is assessed (e.g., recall vs. recognition, short- vs. long-term), or the expected size of the effect. Moreover, without specifying a mechanism (e.g., cortisol-induced disruption of hippocampal function), almost any outcome can be interpreted as consistent with the hypothesis. If memory improves, stress might be said to enhance focus; if it declines, it might be blamed on cognitive overload. This kind of ambiguity impedes falsifiability and enables post hoc rationalization, ultimately undermining theoretical progress⁵⁷. These concerns have prompted growing calls for greater rigor in hypothesis formulation^{56, 58}.”

(6) Please spell out ASN in the header and describe it at first use (it appears in the table on p. 8 before appearing in the main text)

We appreciate the reviewer’s attention to clarity and terminology. We have now spelled out ASN as *Average Sample Number* in the header of Table 4 and introduced the term with a brief explanation in the main text prior to the table on p. 9 (changes highlighted): “For SPRTs and SBFs, where no maximum sample size is specified, simulation-based planning is strongly recommended to estimate the average sample number (ASN) for different effect size scenarios. The ASN refers to the expected number of observations required to reach a decision under a given design

and effect size. Table 3 provides further technical guidance for implementing these simulations using freely available R packages.” This ensures readers unfamiliar with sequential design terminology can follow without confusion.

(7) There appears to be a missing citation (a question mark appears instead) on page 12.

We thank the reviewer for flagging this error. The question mark was an unintended placeholder due to a missing citation key in our reference system. We have now corrected this and inserted the appropriate citation.

(8) Please spell out SBF at first use (p. 12)

We thank the reviewer for pointing this out. We have now spelled out SBF as *Sequential Bayes Factor* at its first mention (now on p. 9) and retained the abbreviation for subsequent uses. This change improves accessibility for readers unfamiliar with the acronym.

(9) TOST is not spelled out (p. 13) or described.

We thank the reviewer for highlighting this oversight. We have now spelled out TOST as the *Two One-Sided Tests* procedure at first mention on p. 14 and briefly described its purpose in equivalence testing (changes highlighted): “*We further ran an equivalence test using the Two One-Sided Tests (TOST) procedure with ± 0.20 as our smallest effect size of interest. TOST evaluates whether the observed effect falls entirely within a predefined equivalence margin by testing two complementary null hypotheses—one that the effect is smaller than the lower bound, and one that it is larger than the upper bound. If both null hypotheses are rejected, the effect is deemed statistically equivalent to zero within the specified bounds. We found that the 90% CI for d was fully contained within these bounds, suggesting the effect of load on RT is practically negligible.*” This clarification helps readers understand its role in evaluating null results and aligns with the general goal of promoting transparent and interpretable statistical reporting.

(10) It may be more straightforward to label Box 3 as just Box 3, rather than Figure 2 and Box 3.

Unfortunately, the double labeling resulted from formatting issues we had with the manuscript which we will fix during typesetting (it is supposed to be a Box but had to be included as the formal type ‘figure’ as we created the box in a different software and later added it to the LaTeX document.

(11) Table 6 is excellent and may be useful to include earlier in the paper. The cognitive psychology aspect of the example tends to get lost in the text. It would also be nice to have additional best practice guidance on describing the result of the

Condition x Time interaction (i.e., which condition increased/decreased more/less over time?).

These are excellent suggestions. We agree that it would be more useful to include Table 6 earlier, so we moved it to the beginning of the paper (now Table 1). We believe that this will not only draw readers' attention towards the specific cognitive psychology example but also be useful as a reference while reading different part of the paper. In addition, we followed the helpful suggestion to add best practice guidance on interpreting the Condition x Time interaction (changes highlighted): "A significant Condition×Time interaction emerged for recall accuracy, $F(1, 54) = 4.37$, $p = .04$, partial $\omega^2 = 0.06$ (95% CI [0.00, 0.23]). Post-hoc simple effects analyses revealed that accuracy decreased significantly from pre- to post-test in the high-load condition (M difference = -0.15, SE = 0.04, $p < .001$, $d = 0.68$), while accuracy remained stable in the low-load condition (M difference = -0.02, SE = 0.04, $p = .63$, $d = 0.09$). This pattern indicates that memory load impaired performance over time, with a medium-to-large effect size for the decline in the high-load condition."

Reviewer #3:

This is an ambitious paper that aims to provide comprehensive recommendations for improving reporting practices in psychological research. The paper presents many valuable points and compiles a wide range of references that will be useful for many researchers. However, I have some reservations, particularly regarding the framing of statistical inference frameworks, which should be addressed before the paper is suitable for publication. Below are some comments, roughly following the order of the paper, though not necessarily reflecting their importance.

We sincerely thank the reviewer for their thoughtful and constructive evaluation of our manuscript. We are grateful for the recognition of our efforts to provide a comprehensive set of recommendations and the acknowledgment that the paper compiles valuable resources for researchers in psychological science.

We also appreciate the reviewer's careful attention to the framing of statistical inference frameworks. We understand the importance of presenting these frameworks accurately and with balanced nuance, and we have carefully revised the relevant sections to address these concerns, as outlined in our detailed, point-by-point responses below.

Major comments

(1) It could perhaps be more illustrative to use a real and not a hypothetical study.

We appreciate the suggestion to use a real empirical study to illustrate best-practice reporting. After careful consideration, however, we prefer to retain the hypothetical cognitive psychology example, which we deliberately chose as the primary worked example in this paper. This approach allows us to demonstrate both insufficient and best-practice reporting side-by-side and avoids anchoring the guidelines to a specific research domain or lab. While we understand the value of real-world examples, the hypothetical example best serves the pedagogical goal of providing a complete, replicable template for transparent reporting.

(2) The presentation of "Null Hypothesis Significance Testing" (NHST) as a well-accepted statistical framework is misleading. In reality, NHST is a confused blend of Fisherian and Neyman-Pearsonian hypothesis testing approaches that is used in practice, but neither Fisher nor Neyman-Pearson advocated it. While Box 1 briefly mentions this, the key conceptual differences could be more stressed:

- In Fisherian testing there is only a null hypothesis H_0 , p-values are interpreted as quantitative evidence against H_0 , H_0 can never be accepted, only rejected or not rejected.

- In Neyman-Pearson testing, there is a null hypothesis H_0 and an alternative hypothesis H_1 , p-values are only a byproduct of a binary decision procedure aimed at controlling the long-run error rates α and β , both H_0 or H_1 can be accepted.

Here are some useful references: Hubbard and Bayarri (2003, 10.1198/0003130031856), Goodman (1993, 10.1093/oxfordjournals.aje.a116700), Goodman (2016, 10.1126/science.aaf5406), Greenland (2019, 10.1080/00031305.2018.1529625), Greenland (2023, 10.1111/sjos.12625), and Gigerenzer (2004, 10.1016/j.socec.2004.09.033), though there are many more.

To avoid reinforcing common misconceptions, these distinctions should be made explicit throughout the paper. For example, the sentence "The statistical power of a test reflects the probability of correctly rejecting the null hypothesis if it is indeed false (which practically means correctly accepting the alternative hypothesis)" conflates Fisherian and Neyman-Pearsonian concepts, contributing to conceptual confusion.

We thank the reviewer for this comment. In response, we have removed the term "NHST" from the manuscript and revised the content of Box 1 accordingly.

Box 1: "Frequentist statistics encompasses two distinct statistical theories that have historically been intermingled and often confused, despite their fundamental and incompatible differences. For further context, see ⁷⁻¹¹, as well as the tutorial by ¹² for pedagogical guidance. The term null hypothesis significance testing (NHST) refers to a hybrid of these two frameworks and is therefore deliberately avoided in this paper.

Significance testing, as developed by Fisher, follows the principle of inductive inference, where conclusions are drawn from the outcome of a particular experiment¹³. In this approach, a single null hypothesis is specified (e.g., that there are no group differences, or that the group difference is not larger than a specific value), and the goal is to assess how unusual the data are if that null hypothesis is true. Knowledge is gained by rejecting this null hypothesis (the hypothesis is "nullified") based on the magnitude of the discrepancy between the observed results and what the null predicts. The p-value is the central statistic in this framework; it quantifies how likely the observed data (or even more extreme outcomes) are, assuming the null hypothesis is true. Researchers may choose different significance thresholds for the p-value depending on the context of their study. A small p-value suggests that the observed data are unlikely under the null hypothesis, leading to the inference that either a rare event has occurred or that the null hypothesis does not adequately explain the data.

The theory of statistical decision making, developed by Neyman and Pearson, follows the principle of inductive behavior ^{14, 15}. In this framework, two exhaustive and mutually exclusive hypotheses are constructed — the null and the alternative

hypothesis. The goal is not to draw inference from a single dataset, but to develop a decision rule that controls the frequency of errors across repeated experiments in the long run. This approach formalizes two types of errors: Type I errors (false positives) and Type II errors (false negatives), and introduces the concept of statistical power — the probability of correctly rejecting the null hypothesis when the alternative is true. Crucially, its purpose is to guide behavior under uncertainty, avoiding the interpretation of single study outcomes as evidence. Decisions are made by setting a fixed criterion (e.g., $\alpha = 0.05$), collecting a sample based on an *a priori* power analysis (e.g., $1 - \beta = 0.95$), and determining whether the test statistic falls within the predefined rejection region (e.g., $p \leq 0.05$). The *p*-value can be used as a test statistic in this context, but it is not considered a measure of evidence. If the test statistic falls within the critical region, the null hypothesis is rejected, which implies a decision to act as though the alternative hypothesis is true. However, since it is still unknown whether this conclusion is correct, the decision reflects a rule for behavior under uncertainty, not a confirmation of the truth of the alternative. In this theory, the same experiment is conceptually repeated an infinite number of times, and only under this repetition do long-run error rates and confidence intervals attain their intended interpretations. For example, a 95% confidence interval means that, across many such repeated experiments, 95% of the intervals would contain the true parameter value.”

We have also rewritten the following section about power on p. 7: “When researchers plan to analyze their data using the Neyman–Pearson frequentist framework, they must first determine their desired statistical power. In this framework, statistical power is the long-run probability of rejecting the null hypothesis when a specific alternative hypothesis is true, and it is given by $1 - \beta^{14}$. As this is a decision-making theory, rejection of the null is interpreted as acting as though the alternative hypothesis is true, without asserting its truth.

Statistical power depends on the statistical test, the planned sample size, the effect size specified under the alternative hypothesis, and the α level. An *a priori* power analysis is therefore conducted before data collection to determine the sample size required to achieve the desired power. If fewer data are collected than planned, the power to detect the specified effect is reduced. If substantially more data are collected, the power increases, making it more likely to detect effects smaller than those originally targeted, which may not be of substantive interest. Consequently, the planned sample size is a critical element of the Neyman–Pearson testing procedure and should align with the study’s theoretical and practical goals.”

(3) The distinction between exploratory and confirmatory research is crucial and should be introduced earlier. The paper appears to focus primarily on confirmatory research, yet in practice, much empirical research is exploratory.

We thank the reviewer for this suggestion and agree that introducing the distinction between confirmatory and non-confirmatory research earlier is very helpful. We therefore moved this section close to the beginning of the manuscript on pp. 4-5, before discussing how to formulate testable hypotheses. In addition, we extended it (changes highlighted):

“When planning a study design and later reporting inferential statistics, one should always distinguish between confirmatory hypothesis testing and non-confirmatory (also known as exploratory) research. Even within purely confirmatory research designs, researchers may conduct additional exploratory analyses, such as assessing the influence of moderators or evaluating different preprocessing choices on a main statistical finding. Such post-hoc analyses should be explicitly labeled as non-confirmatory. Conversely, non-confirmatory research papers can be either purely discovery-oriented or aimed at addressing foundational aspects that strengthen a derivation chain for hypothesis testing^{51, 52}. Such foundational work might include developing and validating psychological measures or experimental manipulations⁵³, examining preprocessing and analysis pipelines⁵⁴, or synthesizing research to establish robust benchmark findings for the field⁵⁵, among other possible topics.

While many parts of this paper focus on confirmatory research designs, many of the reporting principles outlined here, such as transparent documentation of preprocessing decisions, effect size reporting, and clear visualization of results, are equally beneficial for non-confirmatory studies. Transparent reporting in non-confirmatory work helps readers assess the reliability of the findings and facilitates their translation into future confirmatory research.”

In addition, we revised the manuscript to avoid creating the impression that our recommendations are primarily geared towards confirmatory research. For example, in the section on preregistration practices on p. 6, we added the following: “In addition, preregistration can also be a useful tool in exploratory work, where researchers may preregister the planned preprocessing and data analysis steps even if they plan no confirmatory hypothesis tests⁶⁷.”

Moreover, we added a few sentences at the beginning of the “Reporting of results” section regarding exploratory analyses on p. 11 (changes highlighted): “In this section, we give recommendations for how to report findings. We address the most effective ways to visualize descriptive results, how to report statistical results for NHST, Bayesian analyses, sequential designs, and meta-analyses, how to calculate, report, and interpret effect sizes, and how to deal with null findings. Even when analyses are exploratory and do not involve formal hypothesis testing, reporting descriptive statistics, effect sizes, and analytical decisions transparently enhances the interpretability and cumulative value of the research. Labeling analyses as non-

confirmatory or exploratory signals to readers that findings are hypothesis-generating rather than confirmatory.”

Lastly, we added a new paragraph on Hypothesizing After Results are Known (HARKing) to the section on “Common reporting errors and how to avoid them” on p. 17 to discuss a problem particularly prevalent in non-confirmatory research:

“HARKing is a questionable research practice in which post-hoc hypotheses are presented as if they were formulated a priori²³⁹. This may be done deliberately to craft a more compelling or simplified narrative or unintentionally through cognitive biases like hindsight and confirmation bias. Regardless of intent, HARKing misrepresents the research process and can inflate the familywise Type I error rate within a frequentist statistical framework^{229,239}. Exploratory analyses typically involve examining many patterns without prespecified hypotheses. When statistical significance alone determines which results are reported, and no appropriate correction is made for the large number of tests conducted, the risk of false positives increases substantially. This risk is often underestimated because random variability and the ease with which statistical significance can occur in exploratory contexts with many potential hypotheses are not adequately accounted for. Moreover, post-hoc rationalizations can be mistakenly perceived as pre-planned. Preregistration mitigates these issues by requiring clear specification of hypotheses in advance, fostering transparency between exploratory and confirmatory work. Importantly, the problem with HARKing lies not in exploratory research itself, which is vital to discovery, but in misrepresenting it as confirmatory.”

(4) The "Bayesian approach" discussed in the paper seems almost entirely focused on Bayes factor hypothesis testing. For example, the sentence "Bayesian inference, in contrast, evaluates the relative plausibility of hypotheses given the data" suggest this. However, Bayes factor hypothesis testing is only one flavor of Bayesian statistics. In practice, researchers may also use other approaches, such as Bayesian estimation or posterior probabilities.

We thank the reviewer for drawing attention to this important point. We agree that our initial framing could unintentionally give the impression that Bayesian inference is synonymous with Bayes factor hypothesis testing, whereas in reality, Bayesian statistics encompass a broader range of approaches. To address this, we have revised Box 1 and the relevant section on “Reporting inferential statistics” to make clear that Bayesian inference can involve both hypothesis testing using Bayes factors and estimation focused on posterior distributions, credible intervals, and posterior probabilities. We also added sentences highlighting that many Bayesian analyses in psychological research focus on characterizing effect sizes and their uncertainty rather than dichotomous decision-making. Finally, we incorporated citations to widely used sources in the field (McElreath, 2020) to guide readers interested in Bayesian estimation and posterior inference beyond Bayes factors. We

believe these changes clarify the full scope of Bayesian inference and directly address the reviewer's concern.

The changes to Box 1 are highlighted: “Bayesian statistics integrates prior knowledge or beliefs with the observed data through Bayes’ theorem to generate posterior distributions that reflect updated beliefs about parameters or hypotheses¹⁶. Bayesian methods in psychological research can, broadly speaking, serve two complementary purposes. One approach is Bayesian hypothesis testing, most commonly operationalized via Bayes factors, which quantify the relative evidence for H_1 versus H_0 given the data and the specified priors. For example, a Bayes factor BF_{10} of 5 indicates that the observed data are five times more likely under H_1 than under H_0 . This approach is often used for model comparison and, unlike NHST, allows researchers to formally evaluate evidence in favor of the null hypothesis relative to an alternative hypothesis¹⁷.

Another common Bayesian approach is estimation and posterior inference, where the focus lies on characterizing effect sizes and quantifying uncertainty rather than making a dichotomous decision about H_0 . In this framework, posterior distributions, credible intervals, and posterior probabilities provide a nuanced understanding of the data and the likely range of underlying effects¹⁸. Bayesian methods thus offer a flexible inferential framework that encompasses both hypothesis testing and parameter estimation, providing richer information than binary significance testing alone.”

In addition, we added the following section to our recommendations on reporting Bayesian results on p. 12: “Beyond Bayes factor hypothesis testing, many Bayesian analyses in psychological research focus on posterior estimation, which involves summarizing parameters with posterior distributions, credible intervals, and posterior probabilities rather than making dichotomous accept/reject decisions. This estimation-focused perspective can provide a richer understanding of effect sizes and their uncertainty, complementing or even replacing formal hypothesis testing¹⁸.”

We also added a brief sentence on the scope of Bayesian methods discussed in the paper at the very beginning on p. 1: “In our discussion of Bayesian reporting practices, we focus on hypothesis testing using Bayes Factors (BFs) but also cover approaches focused on posterior estimation.”

(5) "Sequential designs" is positioned as an alternative paradigm to Bayesian and frequentist inference, which seems misleading. Frequentist and Bayesian statistics differ in terms of the notion of probability employed (aleatory vs. epistemic). Sequential designs are unrelated to notions of probability but relate to study design. Rather, the counterpart of sequential designs are fixed designs. Both types of designs can be used in frequentist and Bayesian statistics.

We thank the reviewer for this helpful clarification. We agree that our original phrasing could give the impression that sequential designs constitute a third inferential paradigm alongside frequentist and Bayesian approaches, which is conceptually inaccurate. In the revised manuscript, we now clearly describe sequential designs as a study design strategy rather than a statistical framework, and we explicitly note that both sequential and fixed-sample designs can be implemented in either frequentist or Bayesian inference on pp. 1-2 (changes highlighted): “We cover these statistical approaches because they are widely used in psychology and offer distinct advantages. NHST remains the dominant framework, providing well-established significance thresholds but relying on fixed-sample sizes and p-values that are often misinterpreted⁶. In contrast, Bayesian methods offer a probabilistic framework for updating beliefs with data, making them especially useful when prior information is available. In our discussion of Bayesian reporting practices, we focus on hypothesis testing using Bayes Factors (BFs) but also cover approaches focused on posterior estimation. Beyond the choice of inferential framework, researchers must also consider how data collection is planned. This is where sequential designs come into play. Rather than constituting a separate inferential paradigm, a sequential design is a study-planning strategy that can be implemented within either frequentist or Bayesian frameworks. It enhances efficiency by allowing researchers to stop data collection once pre-specified decision criteria are met. Ultimately, both the inferential framework (frequentist vs. Bayesian) and the data collection strategy (fixed-sample vs. sequential) should be chosen based on the study’s goals, the availability of prior knowledge, and practical resource constraints. While the former determines how evidence is quantified and interpreted, the latter dictates when data collection begins and ends.”

We also rewrote parts of the abstract (changes highlighted): “We address considerations across frequentist and Bayesian frameworks and fixed as well as sequential research designs, including guidance on effect size reporting, equivalence testing, and the appropriate treatment of null results.”

(6) Equivalence testing is positioned as a post hoc step following non-significant results, e.g., "In contrast, if the data does not significantly deviate from what is expected under H₀, additional equivalence testing should be conducted before concluding the absence of a theoretically or practically relevant effect or difference" perhaps similar to the "Conditional equivalence testing" approach from Campell and Gustafson (2018, 10.1371/journal.pone.0195145). However, equivalence testing, as originally intended, is a procedure to test the null hypotheses that a parameter is outside an equivalence range (as shown in your Box 3) and this kind of question arises naturally in many studies, e.g., when attempting to "prove" that a new and cheaper treatment is equally good as an established but more expensive treatment, hence the term "equivalence". It could be useful to clarify this. Finally, the sample size

required for achieving satisfactory power in an equivalence test is typically much higher than for a point null hypothesis test, so when following the advocated post hoc equivalence testing approach, the equivalence test will often be underpowered even if the true effect is actually negligible.

We thank the reviewer for this valuable clarification. We agree that our original framing could unintentionally give the impression that equivalence testing is primarily a post hoc tool for interpreting non-significant findings. In the revised manuscript, we now emphasize that equivalence testing was originally developed as a primary inferential approach, particularly in contexts such as bioequivalence or non-inferiority trials, where the central question is whether an effect is small enough to be practically or theoretically negligible. We also highlight that adequate sample size planning is critical, as equivalence tests require larger samples than traditional point-null hypothesis tests to achieve satisfactory power.

While we retain a brief discussion of post hoc equivalence testing as a way to increase the interpretability of null findings, we now explicitly note its limitations, including the bias introduced by the conditional use of equivalence tests and the likelihood of underpowered tests if equivalence testing was not planned in advance. We also added a reference to Campbell and Gustafson (2018) to guide readers toward the “conditional equivalence testing” approach, which formalizes this kind of post hoc procedure. We added this new section on p. 14:

“Equivalence testing is not merely a follow-up to non-significant results, as proposed in the framework of *conditional equivalence testing*¹⁷², but also functions as a primary inferential approach in studies aiming to demonstrate that an effect is sufficiently small to be considered practically or theoretically negligible (e.g., in non-inferiority research). Importantly, reporting equivalence tests only after observing non-significant results can introduce bias by applying different inferential standards to significant versus null findings¹⁷¹. Since equivalence testing requires larger sample sizes to achieve adequate power than traditional significance testing¹⁷³, post hoc applications should be interpreted with caution, particularly when such tests were not pre-registered or planned a priori.”

Minor comments

(1) The term "poor reporting" feels somewhat judgmental. A more constructive alternative might be "incomplete" or "insufficient".

We agree with the reviewer that “poor reporting” may sound overly judgmental. Throughout the manuscript, we replaced this wording with “incomplete” or “insufficient reporting,” which conveys the same meaning in a constructive tone.

(2) Introduction "Given that seeking additional details or clarifications from authors can be challenging and unreliable": Can this statement be substantiated with some references?

We added supporting references that highlight difficulties in obtaining post-publication clarifications from authors and the importance of transparent reporting for reproducibility (Naudet, F. et al. Data sharing and reanalysis of randomized controlled trials in leading biomedical journals with a full data sharing policy: survey of studies published in The BMJ and PLOS Medicine. *BMJ* 360, k400 (2018); Stodden, V., Seiler, J. & Ma, Z. An empirical analysis of journal policy effectiveness for computational reproducibility. *Proceedings of the National Academy of Sciences* 115, 2584–2589 (2018)).

(3) Paragraph "Multiple statistical tests": The section on multiple tests should mention how multiplicity adjustments (e.g., Bonferroni correction) can be incorporated at the design stage to adjust for multiplicity.

We revised the “Multiple statistical tests” paragraph on p. 8 to note that multiplicity adjustments such as Bonferroni or Holm corrections can be pre-specified in the analysis plan to control the family-wise Type I error rate: “When multiple statistical tests are planned, multiplicity adjustments (such as Bonferroni or Holm corrections) can be pre-specified during the design stage to control the family-wise Type I error rate. Incorporating these adjustments into sample size planning ensures that the study remains adequately powered for each test while accounting for the more stringent significance thresholds required to address multiplicity.”

(4) Box 2

- **"exploiting the fact that p-values do not converge under the null hypothesis"**
In this statement, it was not clear to me to "what" p-values should converge under the null? Do the authors perhaps mean that p-values are uniformly distributed under the null? Also, I don't think anything is "exploited".
- **"On average, an SPRT requires 50% fewer data points than a test using a fixed-sample design"**
Over what is the "average" taken? I have doubts that such a general statement can be made as usually such operating characteristics depend on the exact setting (distribution of the data, effect sizes, number of interim analyses, etc.). This sentence needs to be further clarified or toned down.
- **It could be additionally mentioned that sequential designs can be logistically challenging (e.g., the actual costs of the study are not known in advance, interim analyses require unblinding).**

We thank the reviewer for catching these imprecisions. The manuscript has been revised accordingly, as shown in Box 2:

“This restriction is essential because examining the data before reaching the predetermined sample size—and potentially stopping if a statistically significant result is found—can inflate the Type I error rate when p-values are used for inference¹⁹. This inflation occurs because p-values, under the null hypothesis with a continuous and correctly specified null distribution, follow a uniform distribution rather and because p-value inference depends on the stopping rule²⁰, making such interim analysis in fixed designs a questionable research practice known as optional stopping¹⁹.”

“Although SPRT and SBF originate from different statistical paradigms, they are conceptually very similar and both highly efficient. Heuristically, a SPRT can require around 50% fewer data points than a fixed-sample test. However, this efficiency gain depends on the specific combination of the expected and true effect sizes: it tends to be smaller when large effects are expected and can even exceed 50% when small effects are expected^{24,25,31}. A sequential design, however, requires greater flexibility in study logistics than a fixed-sample design, as the final sample size is not known in advance.”

(5) Table 1: Another useful reference for blinded analyses is Dutilh (2021, 10.1007/s11229-019-02456-7)

We added the suggested reference to Table 1 as an additional resource for blinded analyses.

(6) "Bayesian frameworks: Precision and simulation": The section suggests that Bayesian sample size planning always involves simulation, which is not the case. See, e.g., the articles by Wong and Tendeiro (2025, 10.3758/s13428-025-02654-x), Pawel and Held (2025, 10.1080/00031305.2025.2467919), and Pawel et al. (2023, 10.1037/met0000604), which could be mentioned.

This is an excellent point. We clarified that Bayesian sample size planning can involve either simulation-based approaches or analytical solutions on p. 8: “Bayesian sample size determination can be performed using either simulation-based or analytical approaches, depending on the complexity of the model and the planning criterion. For simpler or conjugate models, analytical methods can determine the required sample size to achieve a target posterior precision or a pre-specified Bayes factor without extensive simulations¹⁰⁷⁻¹⁰⁹. In more complex or non-conjugate settings, simulation remains a flexible approach.”

(7) Table 3 could also mention the BFDA package for the <https://github.com/nicebread/bfda%3e>

We added the BFDA package (<https://github.com/nicebread/bfda>) to Table 3 as a resource for planning sequential Bayes factor tests.

(8) The acronym "ASN" is used before being formally introduced, this should be corrected.

We now introduce “average sample number (ASN)” before its first use in the sequential designs section to improve clarity.

(9) "This choice depends on the type of data collected, the assumptions of different statistical methods, whether the study follows a frequentist, Bayesian, or sequential approach.": This phrase suggests that these approaches are mutually exclusive. However, in practice, a study may combine approaches (e.g., report both Bayesian and frequentist analyses), which can enhance robustness.

We revised the sentence to clarify that studies may combine approaches (e.g., reporting both frequentist and Bayesian analyses), which can enhance robustness, on p. 10 (changes highlighted): “When planning their sample size and conceptualizing their study, researchers must decide which statistical models and tests best address their research questions. This choice depends on the type of data collected, the assumptions of different statistical methods, and whether the study follows a frequentist, Bayesian, or sequential approach (or a combination thereof). This section discusses best practices for specifying and reporting statistical models.”

(10) While the grateful package is useful for generating citation reports, it may be worth mentioning that simple citations can be obtained using citation(package = "packagename") in base R.

We added this note to the section discussing the grateful package on p. 10 (changes highlighted): “The grateful R package can be used for generating citation reports, which automatically compile references for the R packages used in an analysis¹⁴⁵. Even more simply, citations can be obtained using `\textit{citation(package = "packagename")}` in base R.”

(11) "Alternative methods, such as the scatterplot, raincloud plot and violin plot can be more informative to examine data distributions": Maybe boxplots could also be mentioned as they improve upon barplots?

We agree that it is important to mention boxplots, which are also shown as an improved data visualization approach in Figure 1. We therefore now mention boxplots instead of scatterplots as examples of better figures on p. 11 (changes highlighted): “However, it has repeatedly shown that barplots may make it difficult for readers to see the data distribution^{148,149}. Alternative methods, such as the boxplot¹⁴⁹, raincloud plot¹⁴⁸ and violin plot¹⁵⁰ can be more informative to examine data distributions, as illustrated in Figure 1.”

(12) "Exact p-values can be informative for readers, and can aid future meta-scientific work, such as calculating effect sizes for meta-analysis when other

statistics are not available.": While exact p-values facilitate meta-analysis, their most fundamental function is providing a quantitative measure of evidence against the null hypothesis, which should be mentioned.

We thank the reviewer for this thoughtful suggestion. While we agree that exact p-values are useful for transparency and facilitate meta-analytic work, we have chosen not to adopt the framing that their primary function is to provide a quantitative measure of evidence against H_0 . From a pure Fisherian perspective, p-values can provide a measure of evidence against the null hypothesis. However, as widely discussed in the statistical literature, p-values do not directly quantify evidential strength: they indicate the probability of obtaining data as extreme as, or more extreme than, the observed data under the assumption that H_0 and its associated model are true. They do not incorporate alternative hypotheses, do not yield posterior probabilities for H_0 , and are sensitive to the choice of test statistic, stopping rules, and sample size. Moreover, in practice, researchers use a hybrid of Fisherian and Neyman-Pearson approaches, in which p-values do not directly quantify evidential strength. For these reasons, we avoid describing p-values as evidence measures and instead highlight their role in transparent reporting and meta-analytic utility, which aligns with the cautious view advocated in contemporary literature (e.g., Wagenmakers, 2007).

(13) "While NHST remains prevalent, its conventional application has well-known limitations, including its reliance on arbitrary significance thresholds and sensitivity to sample size.": Many of these criticisms stem from misunderstandings of p-values and the conflation of Fisherian and Neyman-Pearsonian ideas, see e.g., Greenland (2019, 10.1080/00031305.2018.1529625). Furthermore, Bayes factor thresholds also rely on arbitrary conventions, so this critique should be presented with more nuance.

We agree that our critique was too one-sided and revised the section to clarify that many criticisms stem from misinterpretations of p-values and the hybrid nature of NHST, and we added that Bayes factor thresholds also rely on conventional cut-offs. In addition, we cited Greenland (2019) as suggested. We incorporated these changes in the revised manuscript on p. 12 as follows: "Frequentist statistics remains the dominant framework in psychology and is primarily concerned with guiding statistical decision-making through long-term error control and inductive inference from experimental data, without incorporating subjective prior information. This focus often leads to subtle and frequently misunderstood interpretations of core concepts such as p-values and confidence intervals¹⁵⁴. For example, p-values are commonly misinterpreted as the probability that a hypothesis is true. Bayesian methods offer a complementary approach by explicitly incorporating prior information and enabling direct quantification of evidence for competing hypotheses."

(14) Equivalence testing: It would be important to say that not just any confidence interval can be used but one should use an $(1 - 2\alpha)100\%$ CI to obtain an equivalence test with type-I error rate at most α (e.g., a 90% CI for a 5% level). Equivalently, one could do two one-sided tests at the margin (the TOST procedure) instead of the CI approach, which could also be mentioned.

We thank the reviewer for these helpful suggestions and added that $(1-2\alpha)100\%$ CIs should be used (e.g., 90% for $\alpha = 0.05$) and mentioned that equivalence can also be assessed via the TOST procedure on p. 14 of the revised manuscript (changes highlighted): “To conduct an equivalence test, researchers must establish equivalence margins based on both theoretical and practical considerations relevant to the field¹⁶⁹. These margins define the range within which the effect size is deemed negligible. The test then determines whether the confidence interval around the observed effect falls entirely within this margin, indicating that the effect is equivalent to zero in a practical or theoretical sense (see the left part of Box 3). To maintain a type - I error rate of at most α , equivalence testing should use a $(1 - 2\alpha) \times 100\%$ confidence interval (e.g., 90% for $\alpha = 0.05$). Alternatively, equivalence can be assessed via the two one- sided tests (TOST) procedure, which provides an equivalent formal test of the null hypothesis that the true effect lies outside the prespecified equivalence bounds¹⁶⁹.”

(15) "it is essential to pre-specify equivalence bounds before data analysis to avoid bias": The authors advocated earlier that equivalence tests should be run post hoc when observing null results, hence in most situations researcher would probably not prespecify the margin as they are expecting a genuine effect a priori. Or would the authors suggest to also specify a margin, even when first performing a point null hypothesis test? A useful reference that discusses such post hoc specification issues may be Campbell and Gustafson (2023, 10.15626/MP.2020.2506)

We thank the reviewer for raising this important point regarding the relationship between pre-specified and post hoc equivalence testing. In the revised manuscript, we now clarify that equivalence margins should ideally be pre-specified, as this prevents bias and maintains the nominal type-I error rate. However, we also acknowledge that in practice, researchers may conduct post hoc equivalence tests following non-significant NHST results to improve interpretability. In such cases, we emphasize that the choice of equivalence margin must be explicitly justified after the fact, and that this approach reduces inferential stringency. We have incorporated a citation to Campbell & Gustafson (2023) to guide readers on best practices for handling post hoc margin specification. These revisions make our recommendations consistent with the practical realities of psychological research while maintaining statistical rigor. We added the changes to the section on equivalence tests on p. 14: “Although pre-specifying equivalence margins is preferred to avoid bias, researchers may also perform post hoc equivalence tests

after non-significant results to aid interpretation. In such cases, the chosen margin should be explicitly justified, and the exploratory nature of the analysis should be acknowledged, as post hoc margin selection reduces inferential stringency¹⁷⁰ and conditional equivalence tests can introduce biases¹⁷¹ (for more details, see Selective application of inferential frameworks below). To enhance transparency, researchers are encouraged to report results across a range of plausible margins, demonstrating that conclusions are not unduly dependent on arbitrary choices. Moreover, equivalence margins should be reported for both significant and non-significant findings. Finally, reports should clearly state the selected margins and whether the confidence interval for the observed effect lies entirely within them, confirming equivalence.”

In addition, we rewrote the subsequent section on pp. 14-15 accordingly to match these recommendations (changes highlighted): “As with conventional analyses, researcher flexibility can influence outcomes in both equivalence testing and Bayesian hypothesis testing. In equivalence testing, pre-specifying equivalence bounds before data analysis improves inferential stringency and prevents their selective specification after the results are known¹⁷⁰. If pre-specification is not feasible, researchers should report results across a range of bounds to demonstrate that conclusions are robust and not unduly influenced by arbitrary decisions.”

(16) Table 4: In my view, all regression models in the Table should also be associated with regression coefficients as effect sizes. Usually, one fits a regression model to "adjust" the effect of a treatment of interest with respect to other variables. In this situations, the estimated treatment coefficient (or a transformation, e.g., exp in logistic regression to obtain an odds ratio) is the effect size of interest and not R². Moreover, some of these effect sizes seem more like goodness of fit measures or information criteria (e.g., R² or BIC) rather than actually quantifying the size of an effect in an interpretable way.

We thank the reviewer for this helpful suggestion. In the revised manuscript, Table 4 now lists regression coefficients (β) or their interpretable transformations such as odds ratios for logistic regression and rate ratios for Poisson models as the primary effect sizes for regression models, including Beta regression, LMMs, and generalized linear mixed models GLMMs. Measures such as R², pseudo-R², ICC, and BIC are now explicitly labeled as complementary model-fit indices rather than effect sizes, to avoid conflating fit with interpretable effect magnitudes. These revisions clarify that effect size reporting in regression models should focus on the estimated coefficients for predictors of interest, while model-fit measures provide additional but distinct information.

(17) "Misinterpretation of p-values": I would say that "trending towards significance" is a minor misinterpretation compared to many more severe misinterpretations and misuses of p-values, e.g., interpreting the p-value as the posterior probability of H₀,

or as the probability that "chance" generated the data. I recommend adding more severe ones, see e.g., Greenland et al. (2016, 10.1007/s10654-016-0149-3).

We expanded the “Misinterpretation of p-values” section on p. 18 to include more severe and common misuses, such as interpreting the p-value as the posterior probability of H_0 or as the probability that “chance” generated the data (changes highlighted): “P-values are frequently misinterpreted, which can lead to erroneous conclusions and undermine cumulative science. A common but relatively minor misinterpretation is describing non-significant results as “trending toward significance.” This kind of conclusion, which implies that the test would have been more significant if more data was collected, can be misleading as the collection of additional data may also lead to p-values “trending away” from significance²⁴⁴. Instead, one can write that a p-value was on the threshold of statistical significance.

More severe and widespread misinterpretations include treating the p-value as the probability that the null hypothesis is true, interpreting it as evidence for the alternative hypothesis in comparison to the null hypothesis, or as the probability that the observed data were generated purely by chance^{6,240}. Such interpretations conflate frequentist and Bayesian concepts and overstate the evidential meaning of p-values. Correctly understood, a p-value represents the probability of observing data at least as extreme as those obtained, assuming that H_0 and its associated model are true. Clear communication of this definition is essential to avoid misinterpretation.”

We also included this new section on pp. 17-18 17: **“Misinterpretation of frequentist confidence intervals**

One of the most common misinterpretations of confidence intervals (CIs) is the belief that a 95% CI means there is a 95% probability that the true parameter lies within the observed interval. This interpretation incorrectly treats the interval as a probabilistic statement about the specific sample. However, in the frequentist framework, CIs reflect properties of the statistical procedure, not the individual sample. A 95% CI means that if we were to repeat the same experiment an infinite number of times, using the same procedure each time, approximately 95% of the resulting intervals would contain the true parameter value. It is therefore a statement about the long-run performance of the method, not about the probability of the parameter being in any single observed interval. Understanding this distinction is essential for accurate interpretation and transparent reporting. While CIs provide useful information about estimate precision and uncertainty, they do not quantify the probability of a hypothesis or parameter being true, nor do they offer direct evidence for or against a hypothesis. For further readings see ^{240, 241}. However, there is an interval that can be interpreted in this probabilistic way: in Bayesian statistics, such an interval can be constructed by making additional assumptions about plausible effect sizes, expressed in the form of a prior distribution. These

intervals are usually referred to as credible intervals. See ²⁴² for a Bayesian tutorial with R code, and ²⁴³ for an examination and recommendations regarding robustness.”

(18) Paragraph on post-hoc power: The paper by Hoenig and Heisey (2001, 10.1198/000313001300339897) could be cited.

We thank the reviewer for this suggestion and added the reference to the paragraph on post hoc power.